# Centrifugal multimaterial 3D printing of multifunctional heterogeneous objects

Jianxiang Cheng[1,2], Rong Wang [1,2], Zechu Sun[1,2], Qingjiang Liu[1,2], Xiangnan He[1,2], Honggeng Li[1,2], Haitao Ye[1,2,3], Xingxin Yang[4], Xinfeng Wei[1,2], Zhenqing Li[1,2], Bingcong Jian[1,2], Weiwei Deng[1,5] & Qi Ge [1,2] ✉

There are growing demands for multimaterial three-dimensional (3D) printing to manufacture 3D object where voxels with different properties and functions are precisely arranged. Digital light processing (DLP) is a high-resolution fast-speed 3D printing technology suitable for various materials. However, multi-material 3D printing is challenging for DLP as the current multimaterial switching methods require direct contact onto the printed part to remove residual resin. Here we report a DLP-based centrifugal multimaterial (CM) 3D printing method to generate large-volume heterogeneous 3D objects where composition, property and function are programmable at voxel scale. Centrifugal force enables non-contact, high-efficiency multimaterial switching, so that the CM 3D printer can print heterogenous 3D structures in large area (up to 180 mm × 130 mm) made of materials ranging from hydrogels to functional polymers, and even ceramics. Our CM 3D printing method exhibits excellent capability of fabricating digital materials, soft robots, and ceramic devices.

Additive manufacturing, also known as 3D printing, is an advanced manufacturing technology to create complex 3D objects for a wide range of applications[1-6]. Beyond the conventional techniques of 3D printing with single material, it is desired to develop multimaterial 3D printing capability to manufacture heterogeneous 3D object where the volumetric elements ("voxels") with different properties and functions can be precisely arranged in 3D space[7-9]. Only a few multimaterial 3D printing systems offer such capability by either selectively ink-jetting multiple micro droplets that are cured through photopolymerization[7,8] or developing a multinozzle printing head that generates continuous multimaterial filaments by high-frequency material switching[9]. However, the diversity of printable materials is constrained by the special rheological requirements of these techniques. The feature size and the size of multimaterial transition zoom are also limited by the manner of selectively depositing materials through printing nozzles (Supplementary Fig. 1).

Digital light processing (DLP) 3D printing is a high-resolution fast-speed additive manufacturing technology that forms 3D structures through digitalized UV irradiations that convert liquid photocurable resin to solid 3D structure. DLP can print a variety of materials ranging from hydrogel[10,11], elastomer[12], rigid polymer[13], metal[4,14], ceramic[4,15] to even functional materials that can be either shape changeable[16,17], electric conductive[18,19] or self-healable[20,21]. Recent efforts have been made to significantly improve DLP's printing resolution[4,14], speed[22] and building size[23]. Despite the recent explorations on realizing multi-material 3D printing capability for DLP[24-30], most of the multimaterial switching process requires direct contact of solid wiper[24-26] or fluidic flow[27-29] onto the printed part which constrains DLP-based multi-material 3D printing to small building size, limited available materials, slow speed, severe material contamination, and low function integration.

Here we report a DLP-based centrifugal multimaterial (CM) 3D printing method to generate large-volume heterogeneous 3D objects with multiple properties and functions through precise control on the spatial arrangement of each material voxel. Using

[1]Shenzhen Key Laboratory of Soft Mechanics & Smart Manufacturing, Southern University of Science and Technology, Shenzhen 518055, China. [2]Department of Mechanical and Energy Engineering, Southern University of Science and Technology, Shenzhen 518055, China. [3]Department of Mechanical Engineering, City University of Hong Kong, Kowloon, Hong Kong SAR, China. [4]School of Electronics and Communication Engineering, Guangzhou University, Guangzhou 510006, China. [5]Department of Mechanics and Aerospace Engineering, Southern University of Science and Technology, Shenzhen 518055, China. ✉e-mail: geq@sustech.edu.cn

the CM 3D printing system, we can directly fabricate a large-volume octet truss structure (155 × 108 × 57 mm) where the white and black units are alternatively arranged in space (Fig. 1a). As shown in the zoomed-in images, the CM 3D printing system is able to realize nearly zero material contamination during the multi-material switching process so that the white and black units could be clearly printed. The CM 3D printing system can print more than two materials. Figure 1b presents a printed octet truss structure consisting of four colors (Supplementary Fig. 2) where the layers of white, black, light green and transparent units are stacked from bottom, and the units with four colors are alternatively placed in the top layer. The zoomed-in images confirm that the transitions between different materials are sharp, and no apparent material contaminations can be found. More importantly, the CM 3D printing system is suitable to print a wide range of materials with distinct properties and functions (Supplementary Fig. 3). Figure 1c presents a printed blood vessel system where the red blood vessels are embedded into a transparent hydrogel matrix. As time proceeds, the red "blood" in the blood vessels gradually diffuse into the matrix. Figure 1d shows a Kelvin foam structure where a soft polymer layer is sandwiched by two hard polymer layers (Supplementary Movie 1). Figure 1e demonstrates a printed Miura-origami sheet where the hard polymer panels are connected by the shape memory (SM) polymer hinges which allows the flat Miura-origami sheet to be programmed to a 3D shape (Supplementary Movie 2). Figure 1f presents a flexible ionic conductive (IC) octet truss consisting of an IC elastomer (ICE) core surrounded by the non-conductive soft polymer part (Supplementary Movie 3). Moreover, the CM 3D printing system is also capable of printing multiple ceramics. Figure 1g demonstrates a two-material Kelvin foam structure. After sintering process, the structure made of ceramic-polymer precursor is converted to a pure ceramic structure

(Young's modulus: 122.37 GPa, Supplementary Fig. 4). As summarized in Fig. 1h, the Young's modulus of the materials (Supplementary Fig. 5) that are used in Fig. 1c–g spans in about seven orders of magnitude.

## Results

### Working principle of CM 3D printing system

Figure 2a illustrates the setup of the large-area CM 3D printing system that adopts "bottom-up" projection approach where digitalized UV light is irradiated from the UV projector, which is placed below the printing platform that moves vertically to control the thickness of each slice. Between the printing platform and UV projector, there is a glass plate that supports two or more polymer resin containers and moves horizontally to deliver a needed resin for the corresponding slice. More importantly, we add a rotating motor that spins the printing platform to remove residual resin sticking on the printed part during multimaterial switching (Supplementary Fig. 6). Figure 2b depicts the procedure to print a two-material octet truss. After a slice of the black part is printed in Step I, the printing platform lifts up from the black resin container (Step II). In Fig. 2c, it can be clearly seen that the black residual resin is sticking onto the printed part. In Step III, the rotating motor spins the printing platform to remove the residual resin. Figure 2d shows that the residual black resin is completely removed due to the centrifugal force. Then, in Step IV, the printing continues to complete the white part. Detailed printing and multimaterial switching processes can be found in Supplementary Movie 4. In contrast, if the spinning is not applied to the printing platform, both the printed structure and resin containers are badly contaminated (Supplementary Movie 5 and Supplementary Fig. 7). Moreover, as shown in Supplementary Fig. 8, the CM 3D printing system can even print multimaterial structures with all the channels are perpendicular to the centrifugal force direction. In

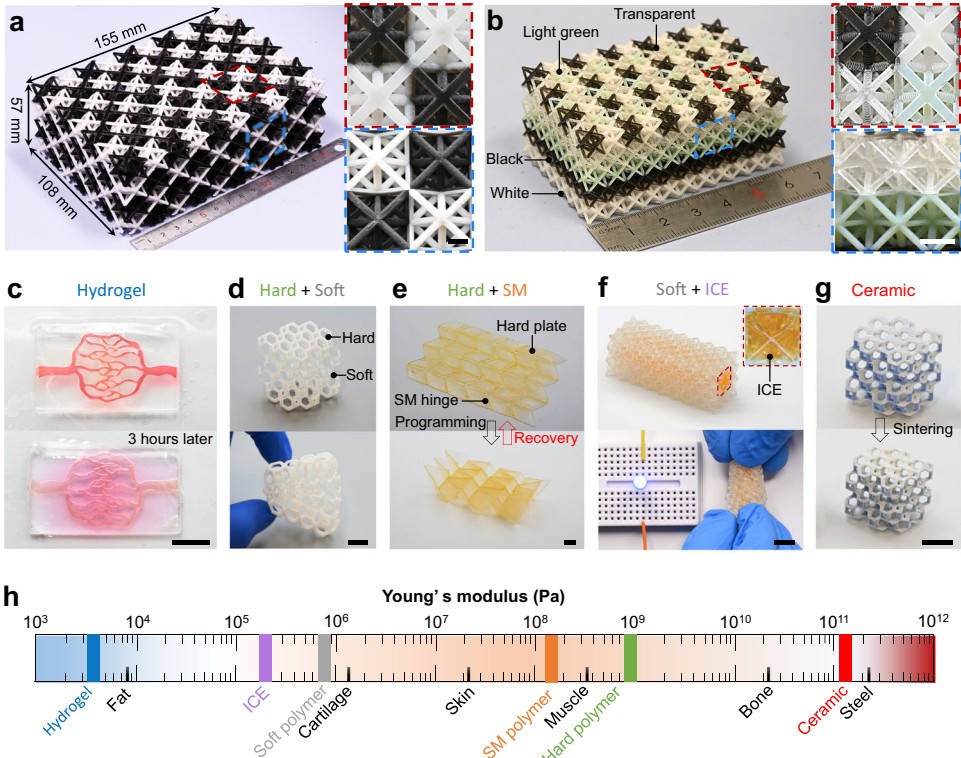

**Fig. 1 | Heterogenous 3D objects created by CM 3D printing system. a** A large-volume two-material octet truss. Scale bars, 10 mm. **b** A four-material octet truss. Scale bars, 5 mm. **c** A blood vessel system consisting of hydrogels with two colors. **d** A Kelvin foam with hard and soft layers. **e** Miura-origami sheet with hard polymer panels and SM polymer hinges. **f** A flexible IC octet truss with an ICE core surrounded by nonconductive soft polymer. **g** A Kelvin foam made of two ceramics. **h** A summary on Young's modulus of the materials that CM 3D printer can print. Scale bars in (**c**–**g**), 10 mm.

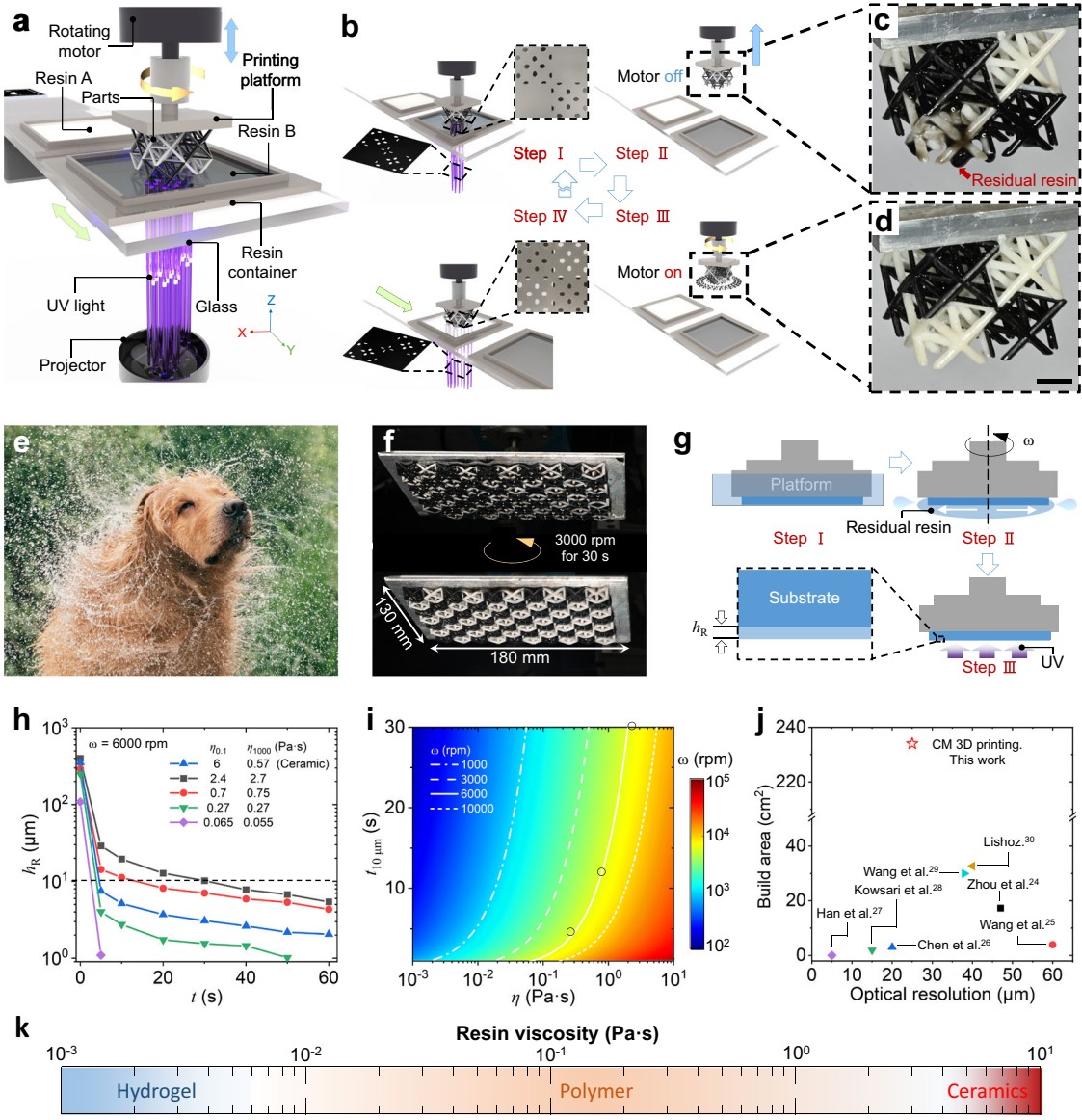

**Fig. 2 | Working principle of CM 3D printing system. a** Illustration of the CM 3D printing system. **b** Steps for multimaterial 3D printing. **c** Residual resin stick onto a printed structure after it leaves the black resin. **d** Residual resin removed by centrifugal force. Scale bars in (**c, d**), 10 mm. **e** Centrifugal force enabled non-contact cleaning inspired by body-shaking of mammals. **f** Large-area cleaning by centrifugal force. **g** Experiment to investigate the effects of spinning time and speed on $h_R$. **h** $h_R$ varies with spinning time for the resins with different $\eta$. **i** Model predictions of $t_{10\mu m}$ for resin with different $\eta$ under different angular speeds. Black circles, experimental data. **j** Comparison on the resolution-build area relation between CM and previously reported 3D printing systems. **k** A summary on viscosity modulus of the resins that CM 3D printer can print.

Supplementary Fig. 9, we schematically illustrate the details on the process of removing residual resin via centrifugal force.

Inspired by mammals who dry themselves through body-shaking (Fig. 2e)[31], we develop the CM 3D printing system that removes the residual resin during multimaterial switching by spinning the printed part with a high angular speed ($\omega$ = 1000-10,000 rpm). Compared with previously reported methods[24–30], the method in this work avoids the direct contact between the printed part and the solid wiper[24–26] or fluidic flow[27–29], and thus is applicable to print multimaterial structures with much greater area. As demonstrated in Fig. 2f, the residual resin on a printed part with a large area (180 × 130 mm) can be quickly removed within 10 s by spinning the printing platform with $\omega$ = 6000 rpm. Moreover, the proposed approach can remove residual resins with a wide range of viscosity. In Fig. 2g, we carried out experiments (details can be found in Methods) to investigate the effects of spinning time and speed on the

thickness ($h_R$) of residual resin with viscosity measured at shear rate of 0.1/s ($\eta_{0.1}$) ranging from 0.065 to 6 Pa·s (Supplementary Fig. 10a). $h_R$ decreases dramatically as the spinning proceeds (Fig. 2h). A higher $\omega$ leads to a faster drop in $h_R$ (Supplementary Fig. 11a). In addition, $h_R$ is independent of the initial area of the resin (Supplementary Fig. 11b), and the location where $h_R$ is measured (Supplementary Fig. 11c). We use the needed duration ($t_{10\mu m}$) after which $h_R$ decreases to 10 μm to quantify the difficulty to remove that resin. In general, the resin with lower $\eta_{0.1}$ has shorter $t_{10\mu m}$ (Fig. 2h). However, it should be noted that $t_{10\mu m}$ of the ceramic resin ($\eta_{0.1}$ = 6 Pa·s) is lower than that of the resin with $\eta_{0.1}$ = 0.7 Pa·s. This is because the ceramic resin exhibits non-Newtonian behavior, and its viscosity at 1000/s is 0.57 Pa·s (Supplementary Fig. 10a). Based on the study on the flow of a viscous liquid on a rotating disk[32], we develop a theoretical model that predicts the relation between $t_{10\mu m}$ and $\omega$ for resin with different $\eta$ by the following equation: $t = 0.75\eta\rho^{-1}\omega^{-2}(h^{-2} - h_0^{-2})$,

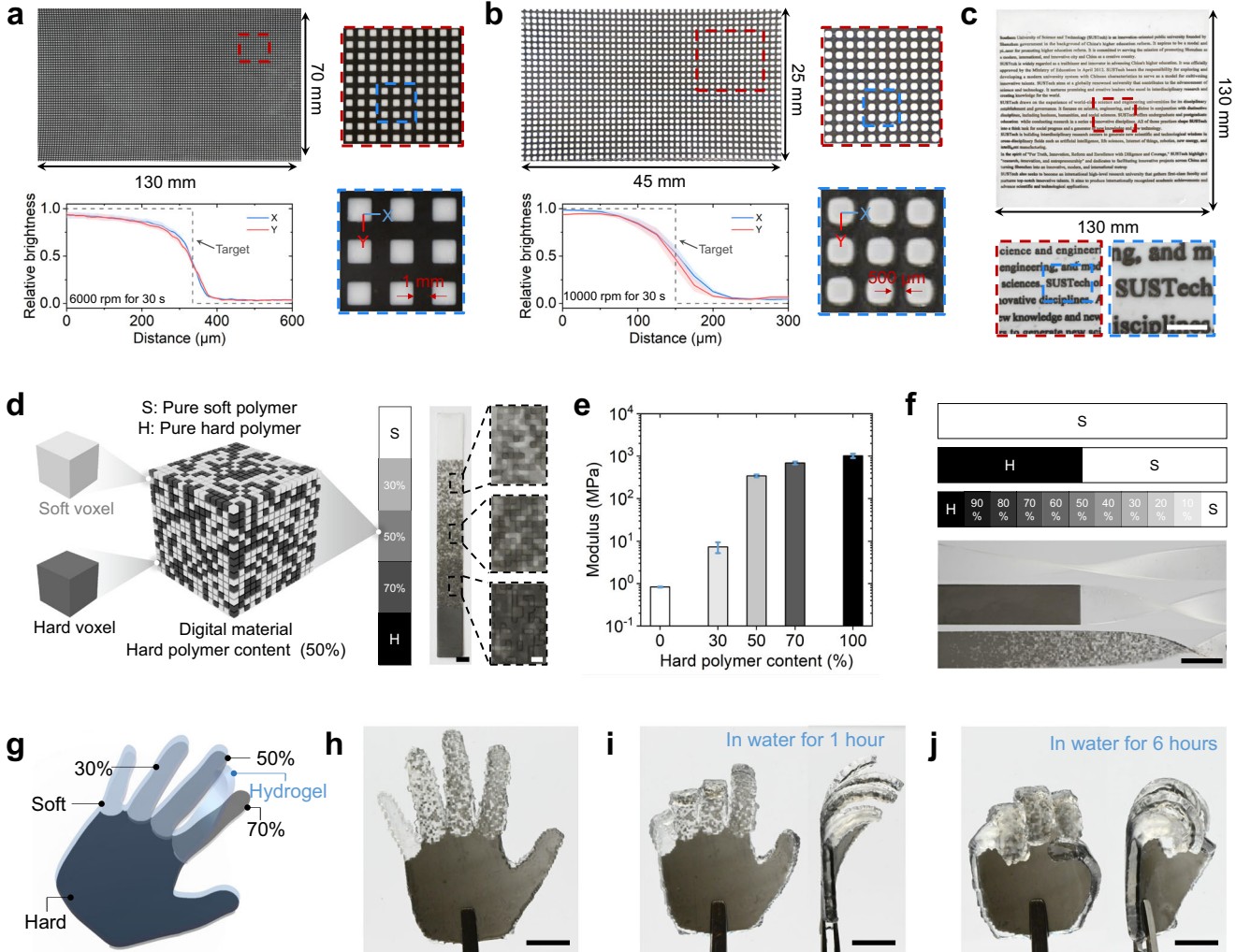

**Fig. 3 | CM 3D printing of digital materials. a**, **b** Grid pattern boards with orthogonal black lines and white squares to investigate the size of two-material transition zoom. **c** A printed letter with black characters embedded into a white board. Scale bars in (**c**), 10 mm. **d** Concept of a printed digital material. Scale bars in (**d**): left, 5 mm; right, 1 mm. **e** Modulus variation of digital material with different

hard polymer content. **f** Demonstration of printed specimens exhibiting multiple mechanical properties at different locations. Scale bars in (**f**), 10 mm. **g**–**j** Demonstrations of a 4D printed hand: design (**g**), printed artifact (**h**), finger bending after being submerged into water for 1 h (**i**) and 6 h (**j**). Scale bars in (**h**–**j**), 10 mm.

where $h_0$ and $h$ are the initial and current thickness of the resin, and $\rho$ is density. Derivation of the equation can be found in Methods. Figure 2i implies that even for a highly viscous resin (10 Pa·s), $t_{10\mu m}$ can be less than 10 s when $\omega = 10^5$ rpm. It should be noted that in the case of printing extremely soft hydrogels (Young modulus: 4 kPa), a high angular speed may lead to severe deformation or even damage of the printed part (Supplementary Movie 6). Thus, a moderate angular speed (less than 3000 rpm) should be used for printing soft hydrogels. In addition, we also conducted experiments to investigate the effect of printed patterns on the efficiency of removing residual resin. As shown in Supplementary Fig. 12 and Supplementary Movie 7, under the same spinning speed and time, the centrifugal force can also efficiently remove the residual resin stick onto complex patterns. In conclusion, the centrifugal force avoids the direct contact to the printed parts during the process of removing residual resin so that the CM 3D printing system can print multimaterial structure with much greater area (Fig. 2j, Supplementary Table 1) and higher printing speed (Supplementary Table 2), and is compatible with a wide range of material resins whose viscosity ranging from $10^{-3}$ to $10^1$ Pa·s (Supplementary Fig. 10b, Supplementary Table 1).

## CM 3D printing of digital materials

To investigate the effect of spinning speed on the transition zoom between two materials, we print grid patterns consisting of orthogonal black lines and white squares. Figure 3a presents a 130 × 70 mm grid pattern board where the width of black line is 1 mm and the distance between neighboring black lines is 2 mm. To print such a large area two-material board, the maximum spinning speed that we could apply to remove residual resin is 6000 rpm above which the printing system shakes violently due to the uneven weight distribution of the printing platform resulted from assembly error. It should be noted that the violent shaking may also be caused during printing a large volume multimaterial structure whose weight is not evenly distributed in horizontal directions. This uneven weight distribution can be balanced by printing extra counter-weight parts (Supplementary Fig. 13). The transition zoom is about 150 μm (Fig. 3a) when a 6000-rpm spinning is applied for 30 s (details on the measurement of transition zoom can be found in Methods). To print a smaller area board with two-material, the maximum spinning speed can be increased to 10,000 rpm which reduces the transition zoom to about 100 μm that is smaller than that from other multimaterial 3D printing techniques (Supplementary Fig. 1). As shown in Fig. 3c, we can print a 130 × 130 mm letter where the

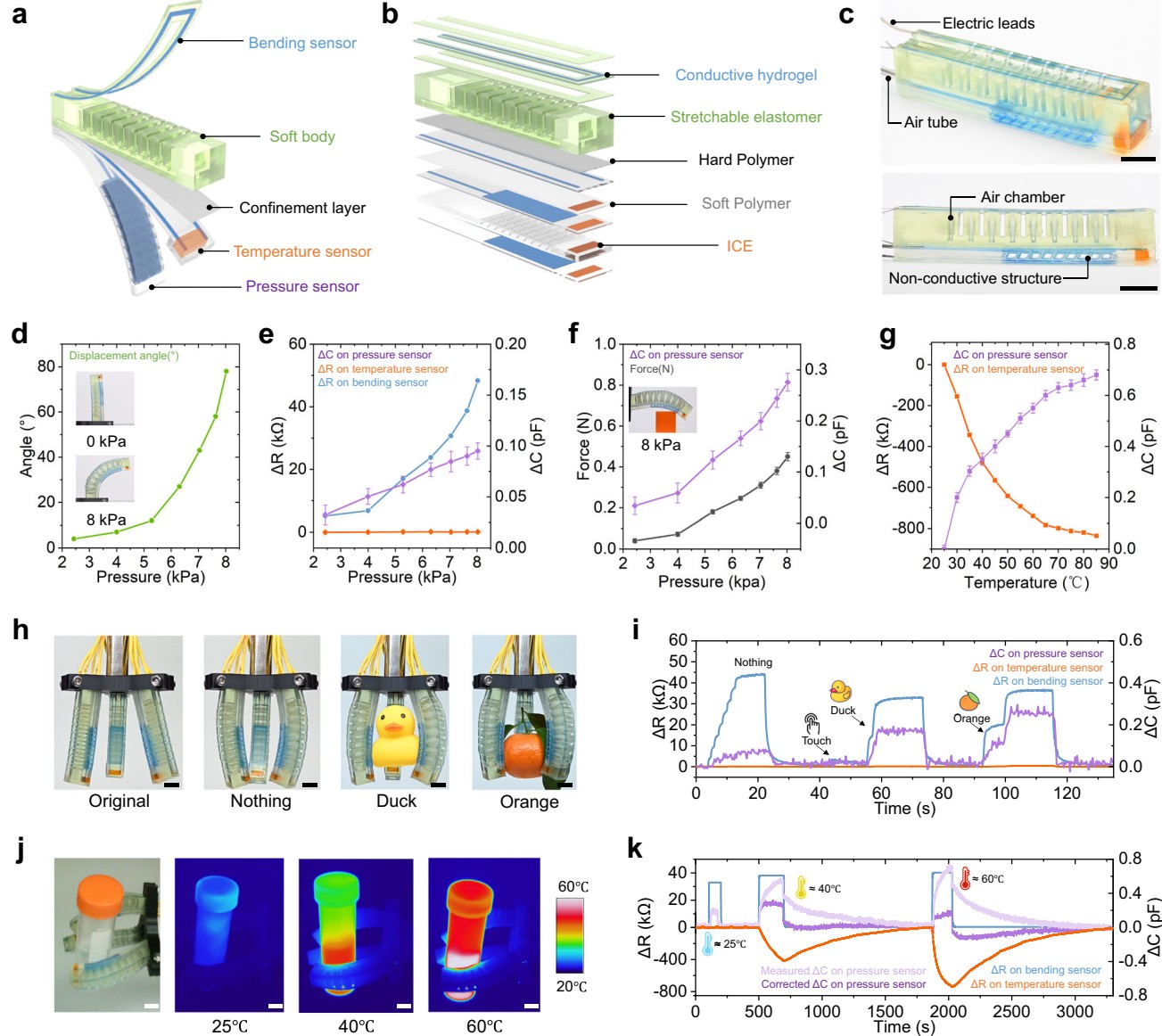

**Fig. 4 | CM 3D printing of soft actuator with multiple sensors. a** Schematic illustration of the SPA with multiple sensors. **b** Materials used to form different parts of the SPA. **c** Snapshots of a printed SPA. **d** Relation between inflation pressure and bending angle of the SPA. **e** Response of three sensors when inflation pressure applied. **f** Response of the pressure sensor when the SPA contacts a rigid obstruction. **g** Response of the temperature and pressure sensors when temperature increases. **h** Snapshots of a soft robotic gripper that grabs different objects. **i**, Readouts of the three sensors when the gripper grabs objects in (**h**). **j** Snapshots of a soft robotic gripper that grabs objects with different temperatures. **k** Readouts of the three sensors when the gripper grabs objects in (**j**). Scale bars in (**c**, **h**, **j**), 10 mm.

black characters are clearly embedded into the white board. The CM 3D printer also enables us to design and fabricate digital materials where the mechanical properties can be tuned by controlling the spatial distribution of the hard and soft voxels (Fig. 3d). By increasing the content of hard voxels from 0 to 100%, the modulus of the printed digital material raises from 0.8 MPa to 1 GPa (Fig. 3e, Supplementary Fig. 14). The capability of printing digital materials allows us to use only two base materials to design and fabricate one single part that exhibits multiple mechanical properties at different locations (Fig. 3f). We further apply this unique capability to a four-dimensional (4D) printing demonstration (Fig. 3g–j) where the palm and five fingers of a hand are formed with different digital materials, and a layer of hydrogel is printed on the top of the hand (Fig. 3g, h). After placing the hand into water for 1 h, the swelling of the hydrogel layer drives the five fingers to bend to different angles due to the different modulus (Fig. 3i). The hand finally makes a fist after being placed into water for 6 h (Fig. 3j).

## CM 3D printing of soft actuator with multiple sensors

The CM 3D printing system enables direct 3D printing a soft pneumatic actuator (SPA) where the bending, pressure and temperature sensors are seamlessly integrated (Fig. 4a). The entire SPA could be fabricated in a single 3D printing with five different polymers including stretchable elastomer, hard polymer, soft polymer, conductive hydrogel, and ICE (Fig. 4b, Supplementary Fig. 15, and Supplementary Table 3). Figure 4c presents the snapshot of the printed SPA with three sensors which connect to electric leads. The SPA bends to 80° upon 8 kPa inflation pressure (Fig. 4d). In Fig. 4e, the bending process leads to an increase in resistance of the bending sensor as well as a slight increase in capacitance of the contact sensor as the bladders of the SPA compress the pressure sensor. In contrast, the resistance of the temperature sensor remains constant. When a rigid obstruction blocks the bending of SPA, the rise in inflation pressure leads to

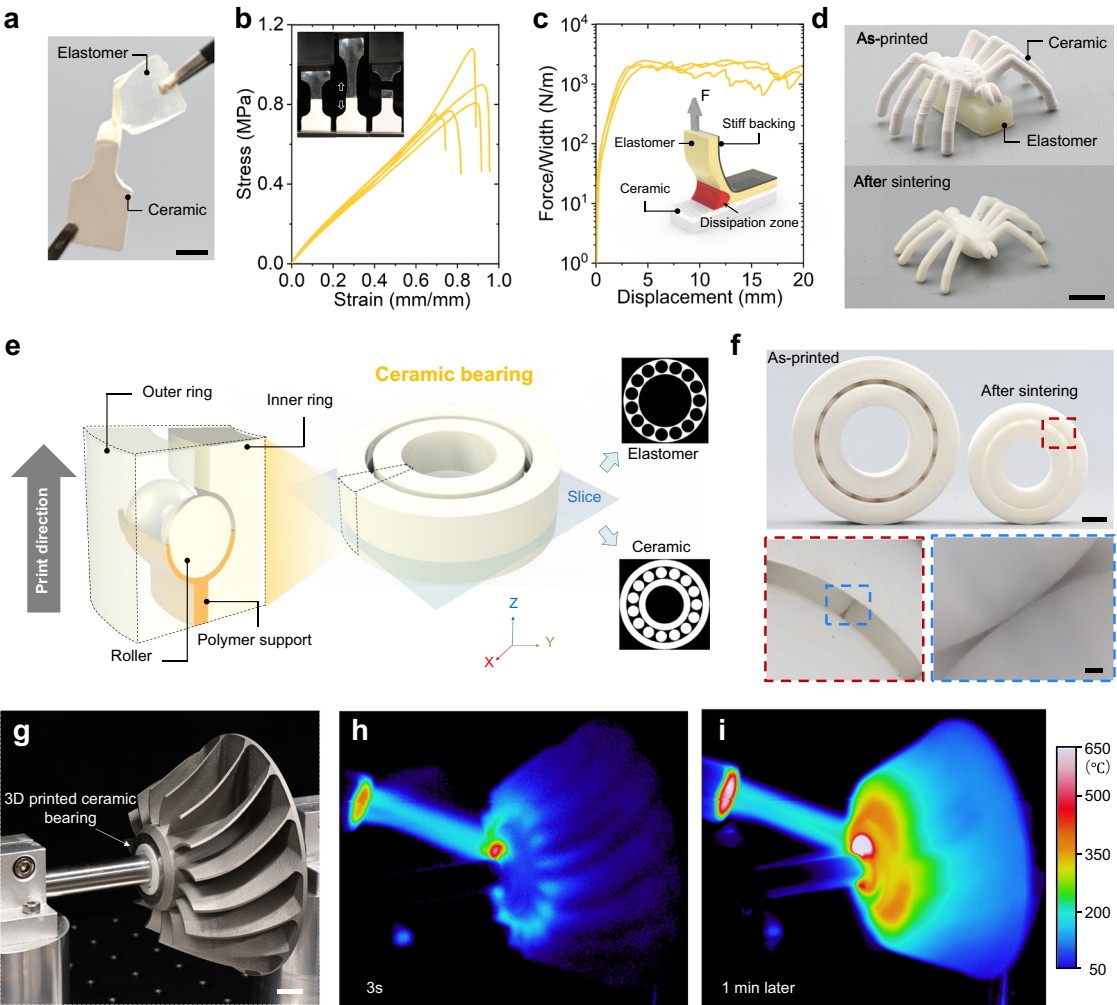

**Fig. 5 | CM 3D printing of ceramic-polymer structures. a** A printed hybrid specimen consisting of ceramic green body and elastomer. Scale bars in a, 5 mm. **b**, **c** uniaxial tensile and 90° peeling tests on the hybrid samples. **d** A printed ceramic spider with overhang body supported by elastomer block. Scale bars in (**d**), 5 mm. **e** Strategy of multimaterial 3D printing a ceramic bearing. **f** Demonstration of a printed ceramic bearing. Scale bars in **f**: top, 5 mm; down, 100 μm. **g** Demonstration of a turbine with a ceramic bearing that connects the metal shaft and impeller. Scale bars in (**g**), 5 mm. **h**, **i** Flame sprayed to impeller and bearing for 3 s (**h**) and 1 min (**i**).

the increase in capacitance of the pressure sensor due to higher contact force applied to the SPA (Fig. 4f). The increase in temperature leads to the decrease in the resistance of the temperature sensor but the increase in the capacitance of the pressure sensor (Fig. 4g). To fully demonstrate the utility of SPA with multiple sensing capabilities, we assembled three SPAs to produce a soft robotic gripper (Fig. 4h). The bending and pressure sensors response differently when the soft robotic gripper grabs nothing, a duck, and an orange, while the resistance of the temperature sensor is constant (Fig. 4i, Supplementary Movie 8). When the robotic gripper grabs a warm or hot object (Fig. 4j), the resistance of the temperature sensor varies correspondingly (Fig. 4k) which could be used to decouple the temperature effect on the pressure sensor (Supplementary Fig. 16).

## CM 3D printing of ceramic-polymer structures
The CM 3D printing system also allows us to print heterogenous 3D structures consisting of ceramic and polymer. We prepare ceramic resin by mixing ceramic particles into acrylate resin which during 3D printing process converts into solid ceramic green body that could form robust interfacial bonding with acrylate elastomer part (Fig. 5a). In a uniaxial test on a hybrid specimen composed of elastomer and ceramic green body arranged in series, the specimen breaks on the elastomer indicating that the interface is stronger than the elastomer (Fig. 5b). The results of the 90° peeling tests confirm that the above conclusion as the measured interfacial toughness is about 1200 J/mm² and interfacial fracture is cohesive (Fig. 5c, Supplementary Fig. 17). Utilizing the strong interfacial bonding between ceramic green body and elastomer, we can print complex ceramic structure with overhang parts. To demonstrate this unique capability, we print a ceramic spider whose body is supported by a solid elastomer part (Fig. 5d). The sintering process removes the elastomer part, and leaves a ceramic spider with an overhang body. To further demonstrate the impact of our approach to manufacture engineering parts, we design a ceramic bearing where there must be empty space between the rollers and the inner/outer ring so that the bearing can rotate freely (Fig. 5e). To support these freestanding rollers, we design and print elastomer to fill the empty space (Supplementary Fig. 18). Sintering process removes the elastomer (Fig. 5f) so that the ball bearing could rotate freely without resistance (Supplementary Movie 9). Figure 5g demonstrates a turbine where the ceramic bearing connects the metal shaft and impeller. The impeller can spin at a high speed due to the low friction of the ceramic bearing (Supplementary Movie 10). The high thermal resistance enables the ceramic bearing to work at 650 °C (Fig. 5h), and its low thermal conductivity prevents the shaft from overheating (Fig. 5i).

## Discussion

We report a DLP-based multimaterial 3D printing approach that utilizes centrifugal force to realize non-contact cleaning of residual resin induced by the multimaterial switching process, and allows us to generate large-volume heterogeneous 3D objects made of materials ranging from hydrogels to functional polymers, and even ceramics. The largest area of a printed two-material structure is 180 × 130 mm, and the lowest width of the two-material transition zoom is about 100 μm. The CM 3D printing system is suitable to print various photopolymers with distinct properties and functionalities. We demonstrate that it is an ideal manufacturing tool to create multimaterial multifunctional structures and devices such as digital materials, soft robot with seamlessly integrated sensors, and ceramic structure with freestanding parts by printing ceramic and polymer together. Our method substantially enhances the multimaterial 3D printing capability for creating multifunctional heterogeneous objects.

## Methods

### Materials

Structures in Fig. 1a, b, Fig. 2c, d, f, and Fig. 3a–c were printed using commercial photo-curable polymer resins including Vero white (white polymer), Vero black (black polymer), Vero clear (transparent polymer), ABS plus (green polymer). Soft polymer resin and elastomer resin in Fig. 1d, f, Fig. 3d–j, Fig. 4b and Fig. 5a, d were printed using commercial photo-curable polymer Agilus. Hard polymer resins in Fig. 1d, e, Fig. 3d–j and Fig. 4b were printed using commercial photo-curable polymer Vero white and Vero clear. All commercial resins were purchased from Stratasys Ltd. (Eden Prairie, MN, USA). Hydrogels in Fig. 1c, Fig. 3g–j, Fig. 4 mainly consist of acrylamide and poly(ethylene glycol) diacrylate (PEGDA)[10]. 1 wt.% red or blue pigment was added to the hydrogel resin for Fig. 1c or Fig. 4. 5 mol/L lithium chloride was added to the blue hydrogel resin in Fig. 4 to achieve ionic conductivity[10]. SM polymer in Fig. 1e consists of 70 wt.% tert- butyl acrylate and 30 wt.% aliphatic urethane diacrylate (AUD)[16]. ICE in Fig. 1f and Fig. 4 mainly consists of butyl acrylate (BA), PEGDA, and lithium chloride (LiCl) with following weight ratios (BA:PEGDA = 98:2, BA + PEGDA: LiCl = 90:10). Stretchable elastomer in Fig. 4 was prepared by mixing ratio of AUD and epoxy aliphatic acrylate[12]. Ceramic resin in Fig. 1g and Fig. 5 was prepared by mixing 1,6-hexanediol diacrylate (Bide Pharmatech Ltd., China), PEGDA and $ZrO_2$ ceramic powders (6.08 g/cm$^3$, $d_{50}$ = 0.56 μm, Shenzhen Adventuretech Co., Ltd., China) with a weight ratio of 4:1:20. Blue ceramic resin in Fig. 1g was modified by adding 1 wt.% $CoAl_2O_4$.

### CM 3D printer

The CM 3D printing system illustrated in Fig. 2a consists of a commercial UV projector (Wintech Digital System Technology Corp, San Marcos, CA, USA), a horizontally moving stage (LTS 150, THROLABS, Newton, NJ, USA) for switching resins, a vertically moving stage (LTS 150, THROLABS, Newton, NJ, USA), and a rotating printing platform that could quickly remove residual resin through centrifugal force (Supplementary Fig. 6a). The printing platform is connected to a rotation motor through a shaft. To ensure that the printing platform can precisely return to the position before spinning, a pair of permanent magnets is equipped to the other end of the shaft, and a pair of permanent magnets with opposite directions are attached to a clamping air cylinder. As illustrated in Supplementary Fig. 6b, after rotating the printing platform to remove residual resin, the two opposite magnets quickly clamp the pair of the magnets attached to the rotating shaft so that the printing platform quickly and precisely return to its initial position. We can adjust the CM 3D printing system into three different configurations for three different printing modes. For the high-resolution-small-area mode, a

Pro4710 Wintech Digital projector is used as the light engine to directly project UV patterns with maximum area of 48 × 27 mm and optical resolution of 25 μm. For the low-resolution-large-area mode, two Pro 6500 Wintech Digital projectors are combined to directly project UV patterns with maximum area of 150 × 160 mm and optical resolution of 75 μm. For the high-resolution-large-area mode, we attach a Pro4710 Wintech Digital projector which projects scrolling images to two orthogonally assembled translational stages (LTS 150, THROLABS) which can quickly move the Pro4710 projector in x and y directions. In this printing mode, the maximum area is 180 × 130 mm, and the optical resolution is 25 μm. Detailed printing process for this mode can be found in Supplementary Movie 4.

### Slicing approach

We design a heterogeneous 3D structure through a commercial computer aided design software (SolidWorks), and save the design model as assembly in.STL format so that different parts are described in the same coordinate system. The STL files for the assembly were then loaded into a self-developed slicing software programmed through MATLAB (MathWorks, Natick, MA, USA). The sliced two 2D images for each layer are arranged in the order that the horizontal stage follows to deliver the resin containers for printing the corresponding parts (Supplementary Fig. 2). When we printed the digital materials in Fig. 3, we generated the bitmaps for the soft and hard parts by randomly arranging the hard pixels with a given composition and the pixels that are not occupied by the hard material are filled by the soft material (Supplementary Fig. 14).

### Measurements on the thickness of the residual resin

We investigated the effects of spinning speed and duration on the thickness of the residual resin on an inversely mounted commercial spin coater (VTC-200 vacuum spin coating machine, HF-Kejing, China) which allows us to accurately control the spinning speed and duration. We chose the Indium Tin Oxide (ITO) glass as the printing substrate due to its low surface roughness, and attached the ITO glass onto the center of the spinning disc of the spinning coater. Before turning on the spinning coater, we deposited a droplet of polymer resin (diameter: ~20 mm, thickness: ~1 mm) onto the ITO glass. The spinning removes most of the residual resin but leaves a thin layer on the ITO glass. In order to measure the thickness of such thin-layer residual resin, we photo-cured it in an oxygen-free environment to eliminate the effect of oxygen inhabitation on the thickness of the cured residual resin. Finally, we measured the thickness of cured residue resin films on a surface roughness measuring instrument (SURFCOM NEX, Tokyo Seimitsu Co., Ltd., Japan).

### Rheological characterization

We measured the viscosity of all polymer resins on Discovery Hybrid Rheometer (DHR2, TA instruments Inc., UK) with a steel plate geometry (diameter: 20 mm). The tests were conducted with shear rate ranging from 0.1 to 1000/s at room temperature. The plate gap was set as 200 μm.

### Measurement on two-material transition zoom

The data of grayscale transition graph was obtained by customized image processing technology (written in Python 3.6, www.python.org, with OpenCV 4.3.0, https://opencv.org). Each raw image photographed by Nikon Z7 under the same lighting and camera parameters. The region of interest (ROI) was obtained through binarization and opening operation (corrosion before expansion). According to the ROI, the sampling lines are equally divided in the X or Y direction. Then the gray value of raw image was traversed and recorded according to each sample line. Finally, all the recorded values are normalized.

## 3D printing and characterization of soft actuator with multiple sensors

We printed the SPA with multiple sensors by following the sequence as shown in Supplementary Table 3. The SPA was printed by using five different materials, and the layer thickness was set to be 100 μm. The variations on the capacitance and resistance of the SPA in Fig. 4 were measured on a precision LCR meter (TH2838H, Changzhou Tonghui Electronic Co., Ltd., China). The testing frequency was 1 kHz.

## 3D printing ceramic-polymer structures

The sintering of the composite structure of ceramic and elastomer was composed of debinding process and sintering process. The debinding process was carried out in a tubular furnace in argon at 800 °C for 2 h to decompose the resin. The sintering process was carried out in a muffle furnace in air at 1450 °C for 2 h. The 90° peeling tests were performed on a MTS universal testing machine (MTS Criterion, Model 43.104 Dimensions, USA) to measure the interfacial toughness between elastomer and ceramic green body.

## Theoretical modeling

We develop a theoretical model that predicts the relation between $t_{10\mu m}$ and $\omega$ for resin with different viscosities based on a previous work[32] but the thin lay of liquid is attached to the bottom surface of the disc (Supplementary Fig. 19a). For simplicity, we make following assumptions: (i) the rotating disc is infinite and horizontal; (ii) the liquid layer is radially symmetric, and extremely thin so that compared with the effect of centrifugal forces, the effect of gravity is negligible; (iii) the liquid is Newtonian so that its viscosity is independent of shear rate; (iv) the radial velocity is so small that Coriolis forces are negligible.

We create cylindrical polar coordinates $(r, \theta, z)$ with the center of the bottom surface of the disk as the origin. The disc spins with a constant angular velocity $\omega$. The initial thickness of the liquid layer is $h_0$ (Supplementary Fig. 19b). Since the liquid is a Newtonian fluid, the shear force $\tau$ along the $z$ direction can be calculated as

$$\tau = \eta \frac{dv}{dz}, \tag{1}$$

where $\eta$ is viscosity, and $v$ is velocity in the radial direction. In Supplementary Fig. 18c, the shear forces on the upper and lower surfaces of an infinitesimal element can be $\tau dS$ and $(\tau + d\tau)dS$ where $dS$ is the surface area of the infinitesimal element.

Based on Supplementary Fig. 19c and d, the centrifugal force acting on the infinitesimal element is:

$$F_c = \rho \omega^2 r dS dz. \tag{2}$$

The total force is balanced in the radial direction:

$$(\tau + d\tau - \tau)dS + \rho \omega^2 r dS dz = 0. \tag{3}$$

By combining Eq. (1) with Eq. (3), we have:

$$-\eta \frac{d^2 v}{dz^2} = \rho \omega^2 r. \tag{4}$$

Equation (4) may be integrated by employing the boundary condition that $\partial v / \partial z = 0$ at the free surface of the liquid ($z = h$) where the shear force must vanish. Thus,

$$\frac{\partial v}{\partial z} = -\frac{1}{\eta} \left[ \rho \omega^2 r z - \rho \omega^2 r h \right]. \tag{5}$$

Equation (5) may be further integrated by employing the boundary condition that $v = 0$ at the surface of the disk ($z = 0$). Thus,

$$v(z) = -\frac{1}{\eta} \left[ \frac{1}{2} \rho \omega^2 r z^2 - \rho \omega^2 r h z \right]. \tag{6}$$

The radial flow $q$ per unit length of circumference is

$$q = \int_0^h v(z) dz = \int_0^h -\frac{1}{\eta} \left[ \frac{1}{2} \rho \omega^2 r z^2 - \rho \omega^2 r h z \right] = \frac{\rho \omega^2 r h^3}{3\eta}. \tag{7}$$

As shown in Supplementary Fig. 19e, the net outflow of the liquid through the infinitesimal element during the time increment $dt$ is:

$$\Delta Q = Q_{out} - Q_{in} = \frac{\partial(rqd\theta)}{\partial r} dr dt = \frac{\partial(rq)}{\partial r} d\theta dr dt \tag{8}$$

As shown in Supplementary Fig. 19f, the net outflow of the liquid leads to the decrease in the volume of the infinitesimal element $\Delta V$:

$$\Delta V = -\frac{\partial h}{\partial t} r d\theta dr dt. \tag{9}$$

Because the net outflow $\Delta Q$ equals to the volume change $\Delta V$, we have:

$$\frac{\partial(rq)}{\partial r} d\theta dr dt + r \frac{\partial h}{\partial t} d\theta dr dt = 0. \tag{10}$$

The above equation can be further simplified as:

$$\frac{\partial(rq)}{\partial r} + r \frac{\partial h}{\partial t} = 0. \tag{11}$$

By using Eq. (7) to substitute $q$, Eq. (11) can be rewritten and reorganized as

$$-2Kh^3 = \frac{\partial h}{\partial t} + 3Krh^2 \frac{\partial h}{\partial r}, \text{ with } K = \frac{\rho \omega^2}{3\eta}. \tag{12}$$

The height variation of any point in the fluid can be obtained by the total derivative expression:[32]

$$\frac{dh}{dt} = \frac{\partial h}{\partial t} + \frac{\partial h}{\partial r} \frac{dr}{dt}. \tag{13}$$

By comparing Eqs. (12) and (13), we can obtain

$$\frac{dh}{dt} = -2Kh^3. \tag{14}$$

By integral of the reorganized form of Eq. (14), we can find the $h$-$t$ relation:

$$-\frac{1}{2h^2} = -2Kt + c. \tag{15}$$

Based on the initial condition $h(t=0) = h_0$, we can find $c$ and reorganize Eq. (15):

$$h = h_0 (4Kth_0^2 + 1)^{-\frac{1}{2}} \text{ or } t = 0.75\eta\rho^{-1}\omega^{-2}(h^{-2} - h_0^{-2}). \tag{16}$$

To calculate the time ($t_{10\mu m}$) after which the thickness of the liquid decreases to 10 μm, we set $h$ to be 10 μm. The initial height of the liquid $h_0$ is set to be 1000 μm.

## Data availability
The XX data generated in this study are provided in the Supplementary Information.

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

## Acknowledgements
We acknowledge the financial supports by the National Key Research and Development Program of China (2020YFB1312900), National Natural Science Foundation of China (No. 12072142), and Science Technology and Innovation Commission of Shenzhen Municipality (ZDSYS20210623092005017).

## Author contributions
Q.G. directed the project. J.C. and Q.G. conceived the idea. J.C. designed and built the multimaterial 3D printer, printed all the 3D structures, and conducted all the experiments. Z.S. developed software code to control the printer. R.W. prepared ceramic solution and performed ceramic sintering. Q.L. and W.D. developed a theoretical model. X.H. prepared hydrogel and ICE solutions. H.L., H.Y. and B.J. helped the structural design. X.Y. measured the two-material transition zoom. X.W. developed the multimaterial slicing software. Z.L. measured the data of SPA sensors. Q.G. drafted the manuscript. J.C., R.W., Q.G. revised the manuscript.

## Competing interests
A Chinese patent on this research has been granted (patent number: ZL202110099819.8). A US patent has been applied by Southern University of Science and Technology on this research (application number: US17/230,616).
