## [Peer Review File · Nature Communications]

Centrifugal Multimaterial 3D Printing of Multifunctional Heterogeneous ObjectsREVIEWER COMMENTS

Reviewer #1 (Remarks to the Author):

The topic is timely and very novel. There are no minor/major comments. It is recommended for publication in its current form.

Reviewer #2 (Remarks to the Author):

The authors present a technical solution to a problem that has plagued many multimaterial vat photopolymerization systems to date: that of contamination. Namely, by integrating a centrifugal assembly into their build platform, they can effectively clean parts before integrating a disparate material. With this technical advance, they present an impressive array of multimaterial use cases ranging from multifunctional soft robots to functionally graded materials. The resolution of their system with respect to the boundaries between materials is also quite good, with relatively sharp transition zones as the authors quantify, enabling digital materials. The use cases the authors present are quite compelling and exhaustive, and the precision of the characterization results they present speaks to the potential of their method.

Since the concept of printing with two different materials in multiple vats via a rotational vat is not new, however, the impressive applications they describe can only be attributed to their centrifugal system if they can convincingly demonstrate that the achieved properties are only possible, or at least are significantly improved, through CM. Put another way, to truly show the advantages of their system, the study requires control experiments comparing the efficacy of their centrifugal approach to existing benchmark approaches to multimaterial printing **without** centrifugation. To start, they quantify the resolution of multimaterial boundaries with centrifugation, but what is the achievable multimaterial resolution without? How much less sharp are multimaterial boundaries without their modified system? And how does this ultimately impact the functionality of their e.g. soft robotic grippers? In addition to such binary control experiments, i.e. with or without centrifugation, it would be highly informative to include process parameter studies on the ideal centrifugation speed and duration to achieve the required cleaning. This would help other researchers to implement the paper's apparatus. More attention needs to be paid to these other multimaterial platforms in the introduction, lest the reader is led to believe that this study is the first stereolithography or digital light processing based multimaterial platform.

The authors claim to have achieved essentially zero contamination in their multimaterial platform, which is indeed apparent in the images they show. This represents a significant benefit of their system. However, we wonder whether this absence of contamination is achievable with geometries beyond those presented by the authors. In particular, the geometries that the authors present are all highly latticed. While lattices are indeed an attractive geometry to print, and represents an attractive target application, given much research and industrial attention paid to such architected metamaterials, we

wonder whether this design choice was not also made to facilitate the cleaning process itself of their new system, as the multitude of voids would likely facilitate the expulsion of trapped resin from internal cavities. Would it be more tedious or time consuming to centrifuge resin from eg internal channels? A study quantifying the efficacy of their centrifugation system for a *range* of test geometries, pointing out where difficulties arise, would be highly informative.

Correspondingly, if indeed the efficacy of their multimaterial cleaning system depends upon the geometry of the printed part, it would be helpful if the authors could use this information to develop a more systematic codesign methodology for their novel system. That is, given an arbitrary part geometry, what are the cleaning conditions required? And how may the part itself be designed taking such restrictions into consideration?

In addition to build area, resolution and viscosity range comparisons authors should compare print duration (mm/min) with Via Stratasys J750 or other current methods mentioned to support high speed nature of proposed method.

The authors state this method can accommodate large volume parts ... are there geometrical constraints that must be considered during part design? Do higher centrifugal forces result in part deformation; Is there a limiting z height that cannot be exceeded?

How does the direction of centrifugal force impact residual resin thickness in different part geometries? Will print designs of channels and cavities normal to centrifugal force require varying spin times? Is there variation in residual resin thickness across part geometries?

Typos:

Line 77: "zoom-in images" to "zoomed-in images"

Line 89: "red bloods" to "red blood"

Overall, however, the authors present a very clever solution to overcome a major technical barrier to multimaterial vat photopolymerization printing - contamination - with well-demonstrated application potential. Their approach is highly applicable to ongoing efforts in multimaterial printing across the three generations of VP methodologies, from SLA, DLP, to CLIP.

Reviewer #3 (Remarks to the Author):

In this manuscript, the authors report a DLP-based multimaterial 3D printing technique by employing a rotating printing platform which spins to clear away residual resin between material switches. As a centrifugal force increases with a rotational speed, even highly viscous resins could be removed. They demonstrated multi-material 3D printing using a wide range of materials including hydrogel, elastomer, conductive material, and ceramic. Examples of the centrifugal multimaterial 3D print include a 4D

printed hand, a soft robotic gripper with integrated sensors, and ceramic parts. All of these examples clearly demonstrate the method's significant potential. Although the implementation of the proposed approach and the demonstrations are both good, the scientific value of the work is questionable.

1. Several alternative methods have been used to demonstrate the removal of residual resin for multimaterial 3D printing. Although technically different, using a centrifugal force does not offer essentially different capability. Resins could be taken off by an air blow or a sponge dab. It appears to have critical issues such as a lengthy wait time (~30s) for each material switch, the possibility of the deformation of the object being printed, particularly if it is soft, and a portion of residual resin being trapped if a print geometry is complex. The radial distance plays a significant role in centrifugal force. As a result, the removal of resin in the region close to the rotational center could be not as effective as in the outer area. The authors stated that their system can realize "nearly zero" material contamination. Given the wetting that occurs between a printed part and the resin, it is impossible to completely remove residual resin. Residual resin is always the same material with the part just printed, so the contact angle should be very small. Removal of resin should have been rigorously investigated and quantified. Overall, the manuscript does not provide a thorough analysis of the proposed method.

2. The theoretical model they proposed is based on the assumption that the spinning substrate is flat, which is too simple to reflect the actual material removal process occurring on a part being printed that is very likely to have a rough surface.

3. According to the video provided, their printer uses scanning to achieve printing over a large area. Build area can be expanded indefinitely if translational stages with longer travel were used, and this has nothing to do with the centrifugal multimaterial 3D printing. Therefore, comparison shown in Fig. 2 (j) cannot support the novelty of the presented method. Scanning over a larger area increases printing time. Spinning time is also added for each material switch. However, no discussion on overall printing time and trade-offs is found in the manuscript.

Comment from Reviewer #1

The topic is timely and very novel. There are no minor/major comments. It is recommended for publication in its current form.

Response: we thank the reviewer for taking precious time to review our paper.

Comments from Reviewer #2

Comment 2.1: the authors present a technical solution to a problem that has plagued many multimaterial vat photopolymerization systems to date: that of contamination. Namely, by integrating a centrifugal assembly into their build platform, they can effectively clean parts before integrating a disparate material. With this technical advance, they present an impressive array of multimaterial use cases ranging from multifunctional soft robots to functionally graded materials. The resolution of their system with respect to the boundaries between materials are also quite good, with relatively sharp transition zones as the authors quantify, enabling digital materials. The use cases the authors present are quite compelling and exhaustive, and the precision of the characterization results they present speaks to the potential of their method.

Response: we thank the reviewer for taking precious time to review our paper and giving us the constructive comments and suggestions.

We are glad to know that the reviewer acknowledges the importance of our work as it presents “a technical solution to a problem that plagued many multimaterial vat photopolymerization systems to date: that of contamination”. More importantly, we emphasize here that the impacts and contributions of our work is **beyond addressing the contamination issue** for multimaterial vat photopolymerization systems:

(i) Our CM 3D printing system **greatly expands the printing area** for DLP-based multimaterial 3D printing from less 40 to **234 cm² (13 cm × 18 cm)**.

(ii) Our CM 3D printing system is suitable to print **a wide range of materials with distinct properties and functions** including hydrogels, soft and hard polymers, hard and shape memory polymers, soft polymer and ionic conductive elastomer and ceramics. In terms of mechanical property, the CM 3D printing system can print materials with Young's modulus ranging from 10^3 Pa to 10^{11} Pa and resin viscosity up to 10 Pa·s.

(iii) Our CM 3D printing system enables us to **directly print structures and devices with multifunctionalities**. We design and print digital materials where the mechanical properties can be tuned by controlling the spatial distribution of the hard and soft voxels. We print a soft pneumatic gripper where the bending, pressure and temperature sensors are seamlessly integrated. The capability of 3D printing ceramic and polymer together allows us to print complex ceramic structure with overhang parts. These unique

demonstrations have not been achieved by other DLP-based multimaterial 3D printing systems.

Comment 2.2: Since the concept of printing with two different materials in multiple vats via a rotational vat is not new, however, the impressive applications they describe can only be attributed to their centrifugal system if they can convincingly demonstrate that the achieved properties are only possible, or at least are significantly improved, through CM. Put another way, to truly show the advantages of their system, the study requires control experiments comparing the efficacy of their centrifugal approach to existing benchmark approaches to multimaterial printing *without* centrifugation. To start, they quantify the resolution of multimaterial boundaries with centrifugation, but what is the achievable multimaterial resolution without? How much less sharp are multimaterial boundaries without their modified system? And how does this ultimately impact the functionality of their e.g. soft robotic grippers?

Response: we thank the reviewer for the suggestion. In the revision, we have added Supplementary Video 5 which compares the multimaterial printing with and without centrifugation. It is apparent that without centrifugation, not only the printed multimaterial structures but also the polymer resin containers are badly contaminated (Figure R1, Supplementary Video 5). In addition, by following the reviewer's suggestion, as shown in Figure R2 (Extended Data Fig. 7), we also printed a grid pattern board without centrifugation. It is clear that if centrifugation is not applied, the printed grid pattern is badly contaminated, and the orthogonal black lines and white squares cannot be clearly distinguished. In contrast, when we printed the grid pattern with centrifugation, we can clearly see the orthogonal black lines and white squares. The nearly zero contamination multimaterial 3D printing capability is even more critical for 3D printing multifunctional soft robotic gripper which integrates multiple electrical units. A certain degree of material contamination would lead to short-circuited and dysfunctional pressure, bending and temperature sensors.

Figure R1 (Supplementary Video 5) | Printed multimaterial structure and polymer resin container are badly contaminated without centrifugal force during multimaterial switch.

Figure R2 (Extended Data Fig. 7) | Comparison on printing a grid pattern with/without centrifugal force.

Comment 2.3: In addition to such binary control experiments, i.e. with or without centrifugation, it would be highly informative to include process parameter studies on the ideal centrifugation speed and duration to achieve the required cleaning. This would help other researchers to implement the paper's apparatus.

Response: We thank the reviewer for the suggestion. In the revision, we have added detailed processing parameters in Extended Data Table 4 (Table R1) for printing different parts.

Structure	Material	Layer Thickness	Curing time	Spinning speed	Spinning time
Fig. 1a, Fig. 2b,c, d, f and Fig. 3a,b,c (polymer with polymer)	Vero black	200 μm	7 s	6000 rpm	30 s
	Vero white	200 μm	12 s	6000 rpm	30 s
Fig. 1b (polymer with polymer)	Vero black	100 μm	5 s	6000 rpm	30 s
	Vero white	100 μm	3 s	6000 rpm	30 s
	Vero clear	100 μm	3 s	6000 rpm	30 s
	ABS plus	100 μm	3 s	6000 rpm	30 s
Fig. 1c (Hydrogel with polymer)	Hydrogel	100 μm	6 s	3000 rpm	30 s
	Red hydrogel	100 μm	6 s	3000 rpm	30 s
Fig. 1d (polymer with polymer)	Vero white	100 μm	5 s	6000 rpm	30 s
	Vero clear	100 μm	3 s	6000 rpm	30 s
Fig. 1e (polymer with polymer)	SMP	100 μm	8 s	6000 rpm	30 s
	Vero Clear	100 μm	3 s	6000 rpm	30 s
Fig. 1f (polymer with polymer)	ICE	100 μm	5 s	6000 rpm	30 s
	Agilius	100 μm	3 s	6000 rpm	30 s
Fig. 1g (ceramic with ceramic)	Ceramic	50 μm	10 s	6000 rpm	30 s
	Blue ceramic	50 μm	10 s	6000 rpm	30s
Fig. 3a, b, c (polymer with polymer)	Vero black	100 μm	5 s	6000 rpm	30 s
	Vero white	100 μm	3 s	6000 rpm	30 s
Fig. 3d, f (polymer with polymer)	Agilius	100 μm	3 s	6000 rpm	30 s
	Vero black	100 μm	5 s	6000 rpm	30 s
Fig. 3h, I, j (polymer with polymer)	Hydrogel	100 μm	6 s	3000 rpm	30 s
	Agilius	100 μm	3 s	6000 rpm	30 s
	Vero black	100 μm	5 s	6000 rpm	30 s
Fig. 4c, d, f, h, j (polymer with hydrogel)	Hydrogel	100 μm	6 s	3000 rpm	30s
	Elastomer	100 μm	5 s	6000 rpm	30 s
	Vero clear	100 μm	3 s	6000 rpm	30 s
	Agilius	100 μm	3 s	6000 rpm	30 s
	ICE	100 μm	5 s	6000 rpm	30 s
Fig. 5a, b, d, f, g, h, i (polymer with ceramic)	Agilius	100 μm	3 s	6000 rpm	30 s
	Ceramic	50 μm	10 s	6000 rpm	30 s

Table R1 (Extended Date Table 4) | Printing parameters for printing all the structures in this work.

Comment 2.3: More attention needs to be paid to these other multimaterial platforms in the introduction, lest the reader is led to believe that this study is the first stereolithography or digital light processing based multimaterial platform.

Response: we thank the reviewer for the comment. In fact, we have conducted a thorough review on the existing DLP-based multimaterial 3D printing platforms in the last sentence of the second paragraph which is highlighted in yellow color in the revision:

“Despite the recent explorations on realizing multimaterial 3D printing capability for DLP²⁴⁻³⁰, most of the multimaterial switching process requires direct contact of solid wiper²⁴⁻²⁶ or fluidic flow²⁷⁻²⁹ onto the printed part which constrains DLP-based multimaterial 3D printing to small building size, limited available materials, slow speed, severe material contamination, and low function integration.”

In this sentence, we have also made a conclusion that the existing multimaterial 3D printing approaches are constrained to small building size, limited available materials, slow speed, severe material contamination, and low function integration since the multimaterial switching process requires direct contact of solid wiper or fluidic flow onto the printed part.

In addition, in Figure 2j, we compare building area between our approach and **previously reported approaches**; in Extended Data Fig. 10b, we compare the range of resin viscosity between our approach and **previously reported approaches**.

Therefore, we believe if reading this paper carefully, the reader will not have the impression that this is the first paper that reports DLP-based multimaterial 3D printing.

Comment 2.4: the authors claim to have achieved essentially zero contamination in their multimaterial platform, which is indeed apparent in the images they show. This represents a significant benefit of their system. However, we wonder whether this absence of contamination is achievable with geometries beyond those presented by the authors. In particular, the geometries that the authors present are all highly latticed. While lattices are indeed an attractive geometry to print, and represents an attractive target application, given much research and industrial attention paid to such architected metamaterials, we wonder whether this design choice was not also made to facilitate the cleaning process itself of their new system, as the multitude of voids would likely facilitate the expulsion of trapped resin from internal cavities. Would it be more tedious or time consuming to centrifuge resin from e.g. internal channels? A study quantifying the efficacy of their centrifugation system for a *range* of test geometries, pointing out where difficulties arise, would be highly informative.

Response: we thank the reviewer for raising this important question. However, we believe the high efficiency of removing the residual resin in our CM 3D printing system is not dependent on the geometry of printed structure.

In this work, the demonstrated multimaterial 3D structures are not limited to latticed structures. **We also printed a plenty of structures which are not latticed, and even include internal channels, but it can be clearly seen that material contaminations are efficiently avoided during the printing of these structures.** Figure R3a (Fig. 1c in the manuscript) presents the printed blood vessel system where the red blood vessels are embedded into the transparent hydrogel matrix which multiple internal channels. Figure R3b (Fig. 3c in the manuscript) shows the printed letter where the black characters embedded into a white board. Figure R3c (Fig. 4d and h in the manuscript) demonstrates digital materials where the black rigid voxels and transparent soft voxels are randomly arranged at each layer. Figure R3d (Fig. 5e in the manuscript) presents the printing of a ceramic bearing where in each layer the elastomer part is embedded into the ceramic part. All the above printed multimaterial structures are not latticed structures, but it can be clearly seen that material contaminations are efficiently avoided. Therefore, we can conclude that the high efficiency of residual resin removal is independent on the geometry of printed structure. This conclusion can be further confirmed by the demonstration of a printed multifunctional soft pneumatic actuator. As shown in Figure R4 (Fig. 4a-c in the manuscript and Extended Data Table 3), the soft pneumatic actuator consists of five materials and entire structure is not latticed. More importantly, the transparent pneumatic body includes channels normal to the direction of centrifugal force, but it is apparent that material contaminations are efficiently avoided.

In order to further convince the reviewer, as shown in Figure R5a (Extended Data Fig. 8a), we also printed the black-white structure where the channels are normal to the direction of centrifugal force. As shown in Figure R5b (Extended Data Fig. 8b), our CM 3D printing system is still able to print such a structure with no obvious material contamination. In Figure R6 (Extended Data Fig. 9), we have added schematic illustrations to explain the reason why our CM 3D printing system can remove the residual resin on the structure where the channels are normal to the direction of centrifugal force. Although the vertical channels are not connected, the residual resin is a continuum and not isolated in each channel. Upon the application of centrifugal force, the residual resin is removed as a whole, and no small portion of residual resin would be trapped in the channels. In addition, as shown in Figure R7 (Extended Data Fig. 12, Supplementary Video 7), we also compare the efficiency of removing residual resins that are stuck onto the printed structure with different patterns. It can be clearly seen under the same spinning condition (speed: 3000 rpm, time: 30 s), the residual resins on all the printed patterns can be quickly removed.

Figure R3 | Summary of the multimaterial structures printed in this work that are not latticed and even include internal channels. a, multimaterial printed blood vessel system (Fig. 1c in the manuscript). **b,** A printed letter with black characters embedded into a white board (Fig. 3c in the manuscript). **c,** Multimaterial printed digital materials (Fig. 4d and h in the manuscript). **d,** Strategy of multimaterial 3D printing a ceramic bearing (Fig. 5e in the manuscript).

Figure R4 (Fig. 4a-c in the manuscript and Extended Date Table 3) | Soft actuator with multiple sensors printed by our CM 3D printing system. a, Schematic illustration of the SPA with multiple sensors. b, Materials used to form different parts of the SPA. c, Snapshots of a printed SPA. d, Detailed process of printing the soft actuator.

Figure R5 (Extended Data Fig. 8) | Printing a multimaterial structure with channels perpendicular to the centrifugal force directions. a, CAD model of the multimaterial structure. b, Snapshot of the structure printed via the CM 3D printing system.

Figure R6 (Extended Data Fig. 9) | Detailed steps to print a multimaterial structure which has two-material parts at each layer and internal channels perpendicular to the centrifugal force directions.

Figure R7 (Extended Data Fig. 12) | Effect of printed patterns on the efficiency of removing residual resin via centrifugal force. a, Snapshots of a printed white substrate, and printed white substrates with different black patterns. **b,** Snapshots of the printed structures which were just lifted from a white resin (viscosity: 0.2 Pa·s). **c,** Snapshots of the printed structures where the white resins were removed by applying 3000 rpm spin for 30 s. Video of the experiment can be found in Supplementary Video 7. Scale bars in c, 10 mm.

Comment 2.5: Correspondingly, if indeed the efficacy of their multimaterial cleaning system depends upon the geometry of the printed part, it would be helpful if the authors could use this information to develop a more systematic codesign methodology for their novel system. That is, given an arbitrary part geometry, what are the cleaning conditions required? And how may the part itself be designed taking such restrictions into consideration?

Response: we thank the reviewer for the suggestion. However, based on the conclusion from the response to the last comment, the high efficiency of removing the residual resin in our CM 3D printing system is **not dependent** on the geometry of printed structure. Compared with the geometry of printed structure, **the key processing parameters** for removing residual resins (spin speed and duration) **are more dependent on the viscosity of polymer resin**. The determination of the two key processing parameters can be guided by experiments and model predictions in Figure 2h and i.

Comment 2.6: In addition to build area, resolution and viscosity range comparisons authors should compare print duration (mm/min) with Via Stratasys J750 or other current methods mentioned to support high speed nature of proposed method.

Response: We thank the reviewer for the suggestion. Following your suggestion, we have added Table S2 (Extended Data Table 2) to compare the printing speed to form a two-material layer of our CM 3D printer with other commercial and reported multimaterial 3D printers.

We emphasize here again that our proposed CM 3D printing approach significantly advances the field of multimaterial 3D printing in the following three aspects:

- (i) Our CM 3D printing system **greatly expands the printing area** of DLP-based multimaterial 3D printing which has much higher printing resolution compared with Polyjet and DIW-based 3D printing.
- (ii) Our CM 3D printing system is suitable to print **a wide range of materials with distinct properties and functions** which the previous DLP-based multimaterial 3D printing methods have not been achieved.
- (iii) Our CM 3D printing system enables us to directly **print structures and devices with multifunctionalities** such as high-resolution digital materials, soft pneumatic gripper with multiple sensors, and ceramic-elastomer hybrid structures.

Compared with Polyjet and DIW-based printing technologies, DLP-based printing is more difficult to realize multimaterial printing as the printing process occurs in a liquid environment. Multimaterial printing for DLP needs the switch of liquid environments, which requires longer time to switch resin containers and remove residual resins. Therefore, in general, DLP-based multimaterial 3D printing is slower than Polyjet to

print a two-material layer. However, compared with other DLP-based multimaterial 3D printing methods, **the printing speed of CM 3D printer stands out as it enables multimaterial printing of a much larger area.**

Printing methods	3D Printer	Resolution	Maximum Printing Area	Printing Mode	Speed of printing two materials in a one layer
DLP	Zhou. et al. ²⁴	Optical resolution: 47 μm	48 mm \times 36 mm	Direct Projection	2.88 mm ² /s
	Wang et al. ²⁵	Optical resolution: 60 μm	26 mm \times 15 mm	Direct Projection	0.65 mm ² /s
	Chen et al. ²⁶	Optical resolution: 20 μm	20 mm \times 15 mm	Direct Projection	Cannot be estimated
	Han et al. ²⁷	Optical resolution: 5 μm	3 mm \times 1.5 mm	Direct Projection	0.18 mm ² /s
	Kowsari et al. ²⁸	Optical resolution: 15 μm	16 mm \times 12 mm	Direct Projection	12.8 mm ² /s
	Wang et al. ²⁹	Optical resolution: 38 μm	73 mm \times 41 mm	Direct Projection	Cannot be estimated
	Lithoz et al. ³⁰	Optical resolution: 40 μm	76 mm \times 43 mm	Direct Projection	Cannot be estimated
	CM 3D Printer in this work	Optical resolution: 25 μm	48 mm \times 27 mm 180 m \times 130 mm	Direct Projection Direct Projection + Scanning	10.8 mm ² /s 39 mm ² /s
	Optical resolution: 75 μm	150 m \times 160 mm	Two-light-engine Projection	200 mm ² /s	
DIW	MM 3D Printer ⁹	Printing Nozzle Diameter: 200 μm	725 mm \times 650 mm	1-Nozzle Printing	2.9 mm ² /s
				8-Nozzle Printing	18.8 mm ² /s
Polyjet	Stratasys J750 ⁸	Build Resolution: +/- 100 μm	490 mm \times 390 mm	-	315 mm ² /s

Table R2 (Extended Data Table 2) | Comparison on the speed of printing two materials in one layer between other multimaterial 3D printers and CM 3D printer in this work.

Comment 2.7: The authors state this method can accommodate large volume parts ... are there geometrical constraints that must be considered during part design? Do higher centrifugal forces result in part deformation; Is there a limiting z height that cannot be exceeded?

Response: we thank the reviewer for raising this interesting question. When we print a large volume part, we do need to consider the geometry of the printed part. However, the geometric aspect we need to consider is **not the geometric complexity** such as whether there are internal channels which would hinder the removal of residual resins, but the planar weight distribution of the printed structure. Since we need to apply spinning on the printing platform, when the printed structure is large but its weight is not well distributed in lateral directions, a high-speed spinning would lead to the violent shaking of the printing system. This violent shaking may also be caused due to the uneven weight distribution of the printing platform resulted from assembly error. For example, as shown in Figure 3a, when we print a large area (130 mm × 70 mm) black-white grid pattern board, the maximum spinning speed that we could apply to remove residual resin is 6000 rpm. Above 6000 rpm, the printing system shakes violently due to the uneven weight distribution of the printing platform resulted from assembly error. In contrast, as shown in Figure 3b, when we print a smaller area (45 mm × 70 mm) grid pattern board, we can apply a spinning speed of 10,000 rpm, the highest speed the current motor can provide.

In conclusion, the uneven planar weight distribution may cause violent shaking of the 3D printing system when the printing platform is spinning at extremely high speed. The uneven planar weight distribution can be caused by assembly error of printing platform which can be addressed by improving the manufacturing and assembling of the printing platform. The uneven planar weight distribution can be caused by the arbitrary geometry of the printed structure. In this case, the problem can be addressed by adding sacrificial parts to balance the uneven weight distribution. In the revision, we have added a few sentences to discuss this problem.

About the question “do higher centrifugal forces result in part deformation?”, the answer is yes. To maintain circular motion of an object, a centripetal force needs to be exerted to the object. The centripetal force can be calculated as $F = m\omega^2r$. The printed part deforms due to the centripetal force. However, the magnitude of such deformation depends on the stiffness of the printed material and structure. As shown in Fig. 2f and Supplementary Video 4, no obvious deformation can be seen when we print a large area (130 mm × 180 mm) lattice structure as the structure made of hard polymers with high Young’s modulus (~ 1 GPa). The centripetal force induced deformation may become an issue when we print extremely soft hydrogel (Young’s modulus: 4 kPa). As shown in Supplementary Video 6, a high angular speed (6000 rpm) may lead to severe deformation or even damage of the printed part. Thus, a moderate angular speed (less than 3000 rpm) should be used for printing soft hydrogels. In fact, the viscosity of

hydrogel solution is less than 10^{-2} Pa·s, based on Fig. 2i, a 1000 rpm spin is sufficient to remove the residual hydrogel solution. In the revision, we have added a few sentences to discuss this problem.

Comment 2.8: How does the direction of centrifugal force impact residual resin thickness in different part geometries? Will print designs of channels and cavities normal to centrifugal force require varying spin times? Is there variation in residual resin thickness across part geometries?

Response: we thank the reviewer for raising the question. Based the response to **Comment 2.4**, we conclude that the high efficiency of removing the residual resin in our CM 3D printing system is not dependent on the geometry of printed structure. As shown in Figure R7 (Extended Data Fig. 12, Supplementary Video 7), we compare the efficiency of removing residual resins that are stuck onto the printed structure with different patterns. It can be clearly seen under the same spinning condition (speed: 3000 rpm, time: 30 s), the residual resins on all the printed patterns can be quickly removed. In Figure R6 (Extended Data Fig. 9), we have added schematic illustrations to explain the reason why our CM 3D printing system can remove the residual resin on the structure where the channels are normal to the direction of centrifugal force. As shown in Figure R6, although the vertical channels are not connected, the residual resin is a continuum and not isolated in each channel. Upon the application of centrifugal force, the residual resin is removed as a whole, and no small portion of residual resin would be trapped in the channels.

Comment 2.9: Typos:

Line 77: “zoom-in images” to “zoomed-in images”

Line 89: “red bloods” to “red blood”.

Response: we thank the reviewer for pointing out the typos. In the revision, we have corrected them correspondingly.

Comment 2.10: Overall, however, the authors present a very clever solution to overcome a major technical barrier to multimaterial vat photopolymerization printing - contamination - with well-demonstrated application potential. Their approach is highly applicable to ongoing efforts in multimaterial printing across the three generations of VP methodologies, from SLA, DLP, to CLIP.

Response: we thank the reviewer again for taking precious time to review our paper and giving us the constructive comments and suggestions.

Comment from Reviewer #3

Comment 3.1: In this manuscript, the authors report a DLP-based multimaterial 3D printing technique by employing a rotating printing platform which spins to clear away residual resin between material switches. As a centrifugal force increases with a rotational speed, even highly viscous resins could be removed. They demonstrated multi-material 3D printing using a wide range of materials including hydrogel, elastomer, conductive material, and ceramic. Examples of the centrifugal multimaterial 3D print include a 4D printed hand, a soft robotic gripper with integrated sensors, and ceramic parts. All of these examples clearly demonstrate the method's significant potential. Although the implementation of the proposed approach and the demonstrations are both good, the scientific value of the work is questionable.

Response: we thank the reviewer for taking precious time to review our paper and giving us the constructive comments and suggestions. In following, we will response the reviewer's comments point by point to answer the reviewer's concerns on the scientific value of this work.

Comment 3.2: Several alternative methods have been used to demonstrate the removal of residual resin for multimaterial 3D printing. Although technically different, using a centrifugal force does not offer essentially different capability. Resins could be taken off by an air blow or a sponge dab. It appears to have critical issues such as a lengthy wait time (~30s) for each material switch, the possibility of the deformation of the object being printed, particularly if it is soft, and a portion of residual resin being trapped if a print geometry is complex.

Response: we thank the reviewer for the comment. As discussed in the last sentence of the second paragraph, the multimaterial switching process in the previously reported DLP-based multimaterial 3D printers requires direct contact of solid wiper (such as sponge dab) or fluidic flow (such as air blow) onto the printed part, which **constrains** DLP-based multimaterial 3D printing to **small building size, limited available materials, slow speed, severe material contamination, and low function integration.**" In contrast, the CM 3D printing system in this work uses centrifugal force to remove residual resin, which avoids the direct contact onto the printed structure during material exchange process and enables direct 3D printing of heterogenous 3D structures in large area made of materials ranging from hydrogels to functional polymers, and even ceramics. Compared with previously reported DLP-based multimaterial 3D printing system, Figure R8a (Fig 2j in the manuscript) shows that the CM 3D printing system can print multimaterial structure with much greater area; Figure R8b (Extended Data Fig. 10b, Extended Date Fig. 10b) shows that the CM 3D printing system is compatible with a wider range of material resins whose viscosity ranging from 10^{-3} to 10^1 Pa·s. In addition, we have added to a table (Table R3, Extended Data Table 2) to compare the speed of printing two materials in one layer between other

multimaterial 3D printers and CM 3D printer in this work. Among all the DLP-based multimaterial 3D printing system, as the CM 3D printer can print much larger two-material area, its speed to printing a two material structures in one layer is highest.

We thank the reviewer for the comment on “possibility of the deformation of the object being printed, particularly if it is soft”. As we all know, to maintain circular motion of an object, a centripetal force needs to be exerted to the object. The centripetal force can be calculated as $F = m\omega^2r$. The printed part deforms due to the centripetal force. However, the magnitude of such deformation depends on the stiffness of the printed material and structure. As shown in Fig. 2f and Supplementary Video 4, no obvious deformation can be seen when we print a large area (130 mm × 180 mm) lattice structure as the structure made of hard polymers with high Young’s modulus (~ 1 GPa). The centripetal force induced deformation may become an issue when we print extremely soft hydrogel (Young’s modulus: 4 kPa). As shown in Supplementary Video 6, a high angular speed (6000 rpm) may lead to severe deformation or even damage of the printed part. Thus, a moderate angular speed (less than 3000 rpm) should be used for printing soft hydrogels. In fact, the viscosity of hydrogel solution is less than 10^{-2} Pa·s, based on Fig. 2i, a 1000 rpm spin is sufficient to remove the residual hydrogel solution. Therefore, the large deformation on printed soft structures can be avoided by applying a lower spinning speed. In the revision, we have added a few sentences to discuss this problem.

We thank the reviewer for the comment on “a portion of residual resin being trapped if a print geometry is complex”. The detailed response to this comment can be found from our response to **Comment 3.4**.

Figure R8. Performance comparison between the CM 3D printer and the previously reported DLP-based multimaterial 3D printing systems. a, Comparison on build area (Fig 2j in the manuscript). **b**, Comparison on the viscosity range of polymer resin (Extended Data Fig. 10).

Printing methods	3D Printer	Resolution	Maximum Printing Area	Printing Mode	Speed of printing two materials in a one layer
DLP	Zhou. et al. ²⁴	Optical resolution: 47 μm	48 mm \times 36 mm	Direct Projection	2.88 mm ² /s
	Wang et al. ²⁵	Optical resolution: 60 μm	26 mm \times 15 mm	Direct Projection	0.65 mm ² /s
	Chen et al. ²⁶	Optical resolution: 20 μm	20 mm \times 15 mm	Direct Projection	Cannot be estimated
	Han et al. ²⁷	Optical resolution: 5 μm	3 mm \times 1.5 mm	Direct Projection	0.18 mm ² /s
	Kowsari et al. ²⁸	Optical resolution: 15 μm	16 mm \times 12 mm	Direct Projection	12.8 mm ² /s
	Wang et al. ²⁹	Optical resolution: 38 μm	73 mm \times 41 mm	Direct Projection	Cannot be estimated
	Lithoz et al. ³⁰	Optical resolution: 40 μm	76 mm \times 43 mm	Direct Projection	Cannot be estimated
	CM 3D Printer in this work		Optical resolution: 25 μm	48 mm \times 27 mm	Direct Projection
180 m \times 130 mm				Direct Projection + Scanning	39 mm ² /s
Optical resolution: 75 μm			150 m \times 160 mm	Two-light-engine Projection	200 mm ² /s
DIW	MM 3D Printer ⁹	Printing Nozzle Diameter: 200 μm	725 mm \times 650 mm	1-Nozzle Printing	2.9 mm ² /s
				8-Nozzle Printing	18.8 mm ² /s
Polyjet	Stratasys J750 ⁸	Build Resolution: +/- 100 μm	490 mm \times 390 mm	-	315 mm ² /s

Table R3 (Extended Data Table 2) | Comparison on the speed of printing two materials in one layer between other multimaterial 3D printers and CM 3D printer in this work.

Comment 3.3: The radial distance plays a significant role in centrifugal force. As a result, the removal of resin in the region close to the rotational center could be not as effective as in the outer area. The authors stated that their system can realize “nearly zero” material contamination. Given the wetting that occurs between a printed part and the resin, it is impossible to completely remove residual resin. Residual resin is always the same material with the part just printed, so the contact angle should be very small. Removal of resin should have been rigorously investigated and quantified. Overall, the manuscript does not provide a thorough analysis of the proposed method.

Response: we thank the reviewer for this comment. We agree with the reviewer’s comment that the radial distance plays significant role in centrifugal force. In fact, as shown in Figure R9 (Extended Data Fig. 11c), we measured the thickness of the residual resin after the application of spinning, and found that the thickness is independent on the distance to the center of rotation. This is because the residual resin stick onto the printing platform is a continuum, and the resin molecules deposited at different locations of the platform interact with each other. When the centrifugal force applied, the residual resin is removed as a whole, and no small portion of residual resin would be left on the flat substrate or trapped in the printed channels. Moreover, in Extended Data Fig. 12 and Supplementary Video 7, we conducted experiments to compare the efficiency of removing residual resins that are stick onto the printed structure with different patterns, and found that under the same condition, the residual resins on all the printed patterns can be quickly removed, and **no residual resins are found to be left or trapped in the center of rotation**. Details can be found in the response to Comment 3.4.

Figure R9 (Extended Data Fig. 11c) | Experiments show that h_R is independent of the location where it is measured ($d_{\text{spin center}}$: distance to the spin center).

It is true that it is impossible to completely remove residual resin. Therefore, in the manuscript, we state that our CM 3D printing system can realize “nearly” zero material contamination but **not “absolute” zero** material contamination. Based on experimental results shown in Fig. 2h, the thickness of residual resin can be reduced from 1 mm to less than 10 μm through centrifugal force, and more than 99% of residual resin even with high viscosity (10 Pa·s) can be removed within 30 s. A less than 10 μm thin film residual resin left on the printed part is acceptable, as Figure 3a shows the lateral transition zoom between two materials is $\sim 100 \mu\text{m}$.

We believe that we have conducted rigorous investigation and quantification on removal of resin. As shown in Fig. 2g, we designed an experiment to investigate the effects of spin speed and time on the thickness of residual resin. It should be noted that we chose the flat ITO glass as the substrate based on the following reasons. After spinning, the thickness of residual resin becomes less than 10 μm or even thinner. Therefore, the roughness of the substrate needs to be much lower than 10 μm , otherwise the thickness of the residual resin after spinning cannot be rigorously investigated and precisely quantified. Moreover, in Extended Data Fig. 12 and Supplementary Video 7, we conducted experiments to compare the efficiency of removing residual resins that are stuck onto the printed structure with different patterns, and found that under the same condition, the residual resins on all the printed patterns can be quickly removed. Details can be found in the response to **Comment 3.4**. In addition, we have also developed a theoretical model that predicts the relation between $t_{10\mu\text{m}}$, ω for resin with different η and create a design map in Fig. 2i to guide the user to find an appropriate combination of spinning speed and time for the polymer resin with different viscosities. Therefore, we believe that this manuscript has provided a thorough enough analysis of the proposed method.

Comment 3.4: The theoretical model they proposed is based on the assumption that the spinning substrate is flat, which is too simple to reflect the actual material removal process occurring on a part being printed that is very likely to have a rough surface.

Response: we thank the reviewer for the comment. In the revision, we have added experiments to investigate the effect of printed pattern on the efficiency of removing residual resin. As shown in Figure R10 (Extended Data Fig. 12, Supplementary Video 7), we compare the efficiency of removing residual resins that are stuck onto the printed structure with different patterns. It can be clearly seen under the same spinning condition (speed: 3000 rpm, time: 30 s), the residual resins on all the printed patterns can be quickly removed. In Figure R11 (Extended Data Fig. 9), we have added schematic illustrations to explain the reason why our CM 3D printing system can remove the residual resin on the structure where the channels are normal to the direction of centrifugal force. As shown in Figure R11, although the vertical channels are not connected, the residual resin is a continuum and not isolated in each channel. Upon the application of centrifugal force, the residual resin is removed as a whole, and no small

portion of residual resin would be trapped in the channels.

Figure R10 (Extended Data Fig. 12) | Effect of printed patterns on the efficiency of removing residual resin via centrifugal force. a, Snapshots of a printed white substrate, and printed white substrates with different black patterns. **b,** Snapshots of the printed structures which were just lifted from a white resin (viscosity: 0.2 Pa·s). **c,** Snapshots of the printed structures where the white resins were removed by applying 3000 rpm spin for 30 s. Video of the experiment can be found in Supplementary Video 7. Scale bars in **c**, 10 mm.

Figure R11 (Extended Data Fig. 9) | Detailed steps to print a multimaterial structure which has two-material parts at each layer and internal channels perpendicular to the centrifugal force directions.

Comment 3.5: According to the video provided, their printer uses scanning to achieve printing over a large area. Build area can be expanded indefinitely if translational stages with longer travel were used, and this has nothing to do with the centrifugal multimaterial 3D printing. Therefore, comparison shown in Fig. 2 (j) cannot support the novelty of the presented method. Scanning over a larger area increases printing time. Spinning time is also added for each material switch. However, no discussion on overall printing time and trade-offs is found in the manuscript.

Response: we thank the reviewer for the comment. To achieve large area printing for DLP-based **single material** 3D printing is not challenging, and can be realized by expanding the projection area or using the projection plus scanning method. However, to achieve large area printing for DLP-based **multimaterial** 3D printing is **extremely difficult**, as the previously reported methods (such as using sponge dab or air blow) cannot rapidly and efficiently remove residual resin stick onto a large area structure. The centrifugal force method proposed in this work has efficiently addressed this issue and enable the CM 3D printer to print much larger area of multimaterial 3D structure (Fig. 2j) with wider viscosity range of polymer resin (Extended Data Fig. 10).

About the time of printing multimaterial 3D printing, in fact that using sponge dab or air blow to remove residual resin also takes time which is longer than our method. However, using sponge dab or air blow could remove the residual resin stucked onto the part with area less than 40 cm². In contrast, in our CM 3D printer, the centrifugal force can remove the residual resin stucked onto the part with area up to 234 cm². In the revision, we have added a table (Table R3, Extended Data Table 2) to compare the speed of printing two materials in one layer between other multimaterial 3D printers and CM

3D printer in this work. Among all the DLP-based multimaterial 3D printing system, as the CM 3D printer can print much larger two-material area, its speed to printing a two material structures in one layer is highest.

REVIEWERS' COMMENTS

Reviewer #2 (Remarks to the Author):

The authors present a technical solution that has overcome a major technical barrier to multimaterial vat polymerization printing: contamination with well-demonstrated application potential. With the changes recorded in the primary/supplementary work, we recommend for publication with no further revision needed; however, the paper could be strengthened by considering the following:

The authors mention in their revision how they design to overcome a potentially critical limitation, on page 11, which is very illuminating:

“It should be noted that the violent shaking may also be caused during printing a large volume multimaterial structure whose weight is not evenly distributed in horizontal directions. This uneven weight distribution can be balanced by printing extra counter-weight parts.”

Perhaps I'm missing something -- we don't see an example of such counter weights in their illustrations and pictures -- but it might be useful to add a schematic or a photograph of what such extra counter weights look like, as it is an important point. How do they propose to design these for an arbitrary uneven weight distribution? How much material do these require? Are they akin to traditional support structures? A brief discussion of the relevant physics of rotational bodies would also be illuminating here.

Reviewer #3 (Remarks to the Author):

The additional information that the authors provided is impressive, but the revised manuscript still has many conclusions and claims that are not supported by sufficient scientific evidence.

Removal of residual resin using centrifugal force is definitely interesting, but it is still not clear why it is fundamentally better than other methods when it still requires a spinning time for material change that is much longer than curing time and when it may damage a part being printed with a soft material. There are no quantitative analyses regarding range of permissible process parameters (e.g. spinning speed) for a given materials properties (stiffness, viscosity and such). It is obvious to use high rpm for a rigid material and low rpm for a soft material. Yet, the authors still claim as if their method works for the entire range of material properties (e.g. 10^3 to 10^{11}).

The large build area is due to scanning (which is not new) and has nothing to do with the presented rotating method. For example, a method reported in Zhou, Rapid Prototyp. J., 3, p153 (2013) can also achieve large build area if their resin vat is made bigger. It involves direct contact for material change, but it would be a problem only for a soft material, which is the same for the method presented in this manuscript. There must be a relationship between stiffness of a material, permissible build area (related

to a radial distance from the spinning center) and spinning speed.

The authors assert that material removal is independent of the geometry of printed structures, but it is too early to say it holds true without careful study considering resin contact angle, surface roughness, curvature of the geometry and other relevant factors.

One of the performance comparison plots also compares two irrelevant quantities. Viscosity and optical resolution are two different quantities having no physical tie. This method can process resins with a wide range of viscosity, but processable resin viscosity is not affected nor has impact on optical resolution.

Reviewer #2 (Remarks to the Author):

Comment: The authors present a technical solution that has overcome a major technical barrier to multimaterial vat polymerization printing: contamination with well-demonstrated application potential. With the changes recorded in the primary/supplementary work, we recommend for publication with no further revision needed; however, the paper could be strengthened by considering the following:

The authors mention in their revision how they design to overcome a potentially critical limitation, on page 11, which is very illuminating:

“It should be noted that the violent shaking may also be caused during printing a large volume multimaterial structure whose weight is not evenly distributed in horizontal directions. This uneven weight distribution can be balanced by printing extra counter-weight parts.”

Perhaps I’m missing something -- we don’t see an example of such counter weights in their illustrations and pictures -- but it might be useful to add a schematic or a photograph of what such extra counter weights look like, as it is an important point. How do they propose to design these for an arbitrary uneven weight distribution? How much material do these require? Are they akin to traditional support structures? A brief discussion of the relevant physics of rotational bodies would also be illuminating here.

Response: we thank the reviewer for taking time to review our manuscript and recommending our paper for publication with no further revision.

For the question about the discussion on Page 11 about potential limitation during printing a large volume part, this is our response to the reviewer’s previous comment on the geometric constraints when printing large parts (**Comment 2.7** in the first-round review). In this paper, all the structures we printed are roughly symmetrical about the rotating axis (Figure R1c) or unsymmetrical with small volume (Figure R1b). The shaking during the spinning of the printing platform is negligible. Therefore, in this paper, we did not really design and print counter-weight parts to make the mass center of the total printed parts on the rotating axis.

However, we believe the violent shaking would happen if we print a large-volume multimaterial structure whose mass center is not on the rotating axis (Figure R1d and 1e). To address this issue, we need to design and print additional part to balance to weight (counter-weight part). As demonstrated in Figure R1f, a simple way to design the counter-weight part is to print additional part which makes mass center of the total printed parts on the rotating axis. In the revision, we have added an Supplementary Fig. 13 in the revision to explain the design of the counter-weight part.

Figure R1 (SupplementaryFig. 13 in the revision) | Shaking during the process of printing unsymmetrical parts. a, CM 3D printing system. **b,** Shaking is negligible when printing a unsymmetrical part with small volume. **c,** Shaking is negligible when printing a symmetric part with large volume. **d,** Shaking becomes violent when printing a unsymmetrical part with large volume. **e,** Violent shaking worsens during printing as the mass center of the printed part drifts away from the rotating axis. **f,** Violent shaking can be eliminated by printing extra counter-weight part which makes the mass center of total printed parts on the rotating axis.

Reviewer #3 (Remarks to the Author):

Comment 3.1: The additional information that the authors provided is impressive, but the revised manuscript still has many conclusions and claims that are not supported by sufficient scientific evidence.

Response: we thank the reviewer for taking time to review our manuscript.

Comment 3.2: Removal of residual resin using centrifugal force is definitely interesting, but it is still not clear why it is fundamentally better than other methods when it still requires a spinning time for material change that is much longer than curing time and when it may damage a part being printed with a soft material. There are no quantitative analyses regarding range of permissible process parameters (e.g. spinning speed) for a given material's properties (stiffness, viscosity and such). It is obvious to use high rpm for a rigid material and low rpm for a soft material. Yet, the authors still claim as if their method works for the entire range of material properties (e.g. 10^3 to 10^{11}).

Response: regard to the comment “it is still not clear why it is fundamentally better than other methods”, we have clearly responded to the reviewer’s previous comment (**Comment 3.2** in the first-round review), the multimaterial switching process in the previously reported DLP-based multimaterial 3D printers requires direct contact of solid wiper (such as sponge dab) or fluidic flow (such as air blow) onto the printed part, which constrains DLP-based multimaterial 3D printing to small building size, limited available materials, slow speed, severe material contamination, and low function integration.” In contrast, **the CM 3D printing system in this work** uses centrifugal force to remove residual resin, which avoids the direct contact onto the printed structure during material exchange process and enables direct 3D printing of heterogeneous 3D structures in large area made of materials ranging from hydrogels to functional polymers, and even ceramics. Compared with previously reported DLP-based multimaterial 3D printing system, Fig 2j in the manuscript shows that the CM 3D printing system can print multimaterial structure with much greater area; Supplementary Fig. 10b shows that the CM 3D printing system is compatible with a wider range of material resins whose viscosity ranging from 10^{-3} to 10^1 Pa·s.

Regarding to the comment “when it still requires a spinning time for material change that is much longer than curing time”, all the DLP-based multimaterial 3D printing methods require additional time to remove the residual resin stick onto the printed part no matter it uses air blow, sponge dab, or centrifugal force. Compared with other methods, our centrifugal force multimaterial 3D printing avoids direct contact to the printed part during the process of removing residual resin, therefore enables direct 3D printing of heterogeneous 3D structures in large area made of materials ranging from hydrogels to functional polymers, and even ceramics. As shown in Table R1 (Supplementary Table 2), we compare the speed of printing two materials in one layer

between other multimaterial 3D printers and CM 3D printer in this work. Among all the DLP-based multimaterial 3D printing system, as the CM 3D printer can print much larger two-material area, its speed of printing a two-material structure in one layer is highest.

Printing methods	3D Printer	Resolution	Maximum Printing Area	Printing Mode	Speed of printing two materials in a one layer
DLP	Zhou. et al. ²⁴	Optical resolution: 47 μm	48 mm \times 36 mm	Direct Projection	2.88 mm ² /s
	Wang et al. ²⁵	Optical resolution: 60 μm	26 mm \times 15 mm	Direct Projection	0.65 mm ² /s
	Chen et al. ²⁶	Optical resolution: 20 μm	20 mm \times 15 mm	Direct Projection	Cannot be estimated
	Han et al. ²⁷	Optical resolution: 5 μm	3 mm \times 1.5 mm	Direct Projection	0.18 mm ² /s
	Kowsari et al. ²⁸	Optical resolution: 15 μm	16 mm \times 12 mm	Direct Projection	12.8 mm ² /s
	Wang et al. ²⁹	Optical resolution: 38 μm	73 mm \times 41 mm	Direct Projection	Cannot be estimated
	Lithoz et al. ³⁰	Optical resolution: 40 μm	76 mm \times 43 mm	Direct Projection	Cannot be estimated
	CM 3D Printer in this work	Optical resolution: 25 μm	48 mm \times 27 mm 180 m \times 130 mm	Direct Projection Direct Projection + Scanning	10.8 mm ² /s 39 mm ² /s
	Optical resolution: 75 μm	150 m \times 160 mm	Two-light-engine Projection	200 mm ² /s	
DIW	MM 3D Printer ⁹	Printing Nozzle Diameter: 200 μm	725 mm \times 650 mm	1-Nozzle Printing	2.9 mm ² /s
				8-Nozzle Printing	18.8 mm ² /s
Polyjet	Stratasys J750 ⁸	Build Resolution: +/- 100 μm	490 mm \times 390 mm	-	315 mm ² /s

Table R1 (Supplementary Table 2) | Comparison on the speed of printing two materials in one layer between other multimaterial 3D printers and CM 3D printer in this work.

Regarding to the comment “when it may damage a part being printed with a soft material”, the demonstrations in Figure 1c, Figure 3h, Figure 4c clearly show that our CM 3D printer can successfully print extremely soft material (i.e. hydrogel) without damage. In addition, we have conducted an experiment that shows using a relatively

low speed spin (less than 3000 rpm), the damage on an extremely soft hydrogel (Young's modulus: 4 kPa) can be avoided. In fact, the viscosity of hydrogel solution is less than 10^{-2} Pa·s, based on Fig. 2i, a 1000 rpm spin is sufficient to remove the residual hydrogel solution. In the revision, we have added a few sentences to discuss this problem. In contrast, air blow or sponge dab method which removes the residual resin by directly contacting the printed part may lead to severe deformation or damage when printing extremely soft hydrogels.

Regard to the comment "there are no quantitative analyses regarding range of permissible process parameters (e.g. spinning speed) for a given materials properties (stiffness, viscosity and such).", **we have developed a theoretical model**

$t = 0.75\eta\rho^{-1}\omega^{-2}(h^{-2} - h_0^{-2})$ **which tells us that the time needed to reduce the thickness**

of the residual resin from h_0 to h is highly dependent on viscosity η and density ρ of the polymer resin as well as the spin speed. However, we don't see it has direct relation with material stiffness. Figure 2h and Supplementary Fig. 11 show that we have conducted thorough quantitative analyses to investigate the key parameters on the time required to reduce the thickness of residual resin to less than 10 μm .

We do not understand the reviewer's comment "It is obvious to use high rpm for a rigid material and low rpm for a soft material." Based on our experiments and model prediction (Figure 2h and 2i), the spinning speed or spinning time are highly dependent on the viscosity of the polymer resin rather than stiffness of printed material. In fact, as shown in Supplementary Fig. 11, the ceramics has highest stiffness, but the spin speed to remove its residual resin isn't the highest.

We respectfully disagree the reviewer's comment "Yet, the authors still claim as if their method works for the entire range of material properties (e.g. 10^3 to 10^{11})." Throughout the whole paper, we have demonstrated many printed structures that are made of materials ranging from hydrogels to functional polymers, and even ceramics. We also carried out experiments to measure Young's moduli of these materials (Supplementary Fig. 4 and Fig. 5).

Comment 3.3: The large build area is due to scanning (which is not new) and has nothing to do with the presented rotating method. For example, a method reported in Zhou, Rapid Prototyp. J., 3, p153 (2013) can also achieve large build area if their resin vat is made bigger. It involves direct contact for material change, but it would be a problem only for a soft material, which is the same for the method presented in this manuscript. There must be a relationship between stiffness of a material, permissible build area (related to a radial distance from the spinning center) and spinning speed.

Response: as we responded to the reviewer's previous comment (**Comment 3.5** in the first-round review), to achieve large area printing for DLP-based **single material 3D** printing is not challenging, and can be realized by expanding the projection area or using the projection plus scanning method. However, to achieve large area printing for

DLP-based **multimaterial** 3D printing is **extremely difficult**, as the previously reported methods (such as using sponge dab or air blow) cannot rapidly and efficiently remove residual resin stick onto a large area structure. The centrifugal force method proposed in this work has efficiently addressed this issue and enable the CM 3D printer to print much larger area of multimaterial 3D structure (Fig. 2j) with wider viscosity range of polymer resin (Supplementary Fig. 10).

We respectfully disagree the reviewer’s comment “For example, a method reported in Zhou, Rapid Prototyp. J., 3, p153 (2013) can also achieve large build area if their resin vat is made bigger.”. This is reviewer’s imagination. As listed in Table R1, the printing area which Zhou’s system can achieve is 48 mm × 36 mm. No clue shows that they can achieve a printing area bigger than this value for multimaterial 3D printing.

We also respectfully disagree the reviewer’s comment “There must be a relationship between stiffness of a material, permissible build area (related to a radial distance from the spinning center) and spinning speed.”. The experiments and model predictions show the thickness of the residual resin is dependent on viscosity of polymer resin as well as spin speed and time, but independent on the size of printed structure. There is no direct relation between spin time and material stiffness. Especially, as shown in Figure 2h, removing the residual ceramic resin takes less time than many other polymer resin.

Comment 3.4: The authors assert that material removal is independent of the geometry of printed structures, but it is too early to say it holds true without careful study considering resin contact angle, surface roughness, curvature of the geometry and other relevant factors.

Response: as we responded to the reviewer’s previous comment (**Comment 3.4** in the first-round review), we have added experiments to investigate the effect of printed pattern on the efficiency of removing residual resin. Based on the experimental results, we can conclude the material removal is independent of the geometry of printed structures.

As shown in Figure R2 (Supplementary Fig. 12, Supplementary Video 7), we compare the efficiency of removing residual resins that are stuck onto the printed structure with different patterns. It can be clearly seen under the same spinning condition (speed: 3000 rpm, time: 30 s), the residual resins on all the printed patterns can be quickly removed. In Figure R3 (Supplementary Fig. 9), we have added schematic illustrations to explain the reason why our CM 3D printing system can remove the residual resin on the structure where the channels are normal to the direction of centrifugal force. As shown in Figure R3, although the vertical channels are not connected, the residual resin is a continuum and not isolated in each channel. Upon the application of centrifugal force, the residual resin is removed as a whole, and no small portion of residual resin would be trapped in the channels.

Figure R2 (Supplementary Fig. 12) | Effect of printed patterns on the efficiency of removing residual resin via centrifugal force. a, Snapshots of a printed white substrate, and printed white substrates with different black patterns. **b,** Snapshots of the printed structures which were just lifted from a white resin (viscosity: 0.2 Pa·s). **c,** Snapshots of the printed structures where the white resins were removed by applying 3000 rpm spin for 30 s. Video of the experiment can be found in Supplementary Video 7. Scale bars in **c**, 10 mm.

Figure R3 (Supplementary Fig. 9) | Detailed steps to print a multimaterial structure which has two-material parts at each layer and internal channels perpendicular to the centrifugal force directions.

Comment 3.5: One of the performance comparison plots also compares two irrelevant quantities. Viscosity and optical resolution are two different quantities having no physical tie. This method can process resins with a wide range of viscosity, but processable resin viscosity is not affected nor has impact on optical resolution.

Response: we thank the reviewer for the constructive suggestion. As shown in Figure R4 (Supplementary Fig. 10b in the revision), in the revision, we have used the viscosity-build area plot to replace the viscosity-optical resolution plot.

Figure R4 (Supplementary Fig. 10b) | Comparison on the relation between the viscosity range of polymer resin and build area of different DLP-based multimaterial 3D printing methods.